# The geometry of sea-level change across a mid-Pliocene glacial cycle

Meghan. E. King[1*], Jessica. R. Creveling[1], Jerry. X. Mitrovica[2]

[1]College of Earth, Ocean, and Atmospheric Sciences, Oregon State University, Corvallis, OR 97331, USA
[2]Department of Earth and Planetary Sciences, Harvard University, Cambridge, MA 02138, USA
*Currently at Department of Earth and Space Sciences, University of Washington, Seattle, WA 98195, USA

*Correspondence to:* Meghan E. King (mking5@uw.edu)

**Abstract.** Predictions for future sea-level change and ice sheet stability rely on accurate reconstructions of sea level during past warm intervals, such as the mid-Pliocene Warm Period (MPWP; 3.264 – 3.025 Ma). The magnitude of
MPWP glacial cycles, and the relative contribution of meltwater sources, remains uncertain. We explore this issue by modeling glacial isostatic adjustment processes for a wide range of possible MPWP ice sheet melt zones, including North America, Greenland, Eurasia, West Antarctica, and the Wilkes Basin, Aurora Basin, and Prydz Bay Embayment in East Antarctica. As a case study, we use a series of ice histories together with a suite of viscoelastic Earth models to predict global changes in sea level from the Marine Isotope Stage (MIS) M2 glacial to the MIS
KM3 interglacial. At Whanganui Basin, New Zealand, a location with stratigraphic constraints on Pliocene glacial– interglacial sea level amplitude, the calculated local sea-level (LSL) rise is on average ~15% lower than the associated global mean sea level (GMSL) change of the ice sheet scenarios explored here. In contrast, the calculated LSL rise across the MIS M2 to KM3 deglaciation at Enewetak Atoll is systematically larger than the GMSL change by 10%. While no single LSL observation (field site) can provide a unique constraint on the sources of ice melt
during this period, combinations of observations have the potential to yield a stronger constraint on GMSL change and to narrow the list of possible sources.

## 1 Introduction

Accurate reconstructions of sea level during past warm periods offer insight into ice sheet stability in the face of projected anthropogenic climate change (Dutton et al., 2015). In this regard, the mid-Pliocene Warm Period (MPWP; 3.264 – 3.025 Ma) serves as a key period of focus. Mid-Pliocene reconstructed atmospheric $CO_2$ and global mean annual surface temperatures are comparable to projected 21st century warming scenarios (350-450 ppm and ~2-3°C above modern, respectively; Pagani et al., 2010; Haywood et al., 2013) and, as such, estimates of

Pliocene peak global mean sea level (GMSL) have calibrated the sensitivity of global climate models (DeConto and Pollard, 2016). While the differing rates of $CO_2$ forcing, and the distinct oceanographic conditions from the closing of equatorial seaways (Haywood et al., 2011; Sarnthein et al., 2009), may reveal the MPWP as an imperfect analogue for the future, the mid-Pliocene remains a crucial natural laboratory for evaluating the complexity of Earth's ice age climate system.


A rich literature has sought to quantify GMSL variability during the MPWP using ice sheet modeling (DeConto and Pollard, 2016; de Boer et al., 2017; Berends et al., 2019) and a suite of proxy data, including $\delta^{18}O$ records, with and without complementary Mg/Ca measurements (e.g., Dwyer and Chandler, 2009; Sosdian and Rosenthal, 2009; Rohling et al., 2014; Winnick and Caves, 2015; Miller et al., 2020), phreatic overgrowths on speleothems (Dumitru

et al., 2019), sequence stratigraphic records (e.g., Wardlaw and Quinn, 1991; Naish and Wilson, 2009; Miller et al., 2012; Grant et al., 2019) and coastal plain terraces and escarpments (e.g., Dowsett an Cronin, 1990; Krantz, 1991; Kaufman and Brigham-Grette, 1993; James et al., 2006; Rowley et al., 2013; Rovere et al., 2014; Hearty et al., 2020; Sandstrom et al., 2021). These studies have evaluated either the total amplitude of sea-level change through Pliocene glacial–interglacial cycles or the absolute peak in sea level during the Pliocene 'super-interglacials' yet

have achieved little consensus on these values. It is common within these studies to infer the suite of ice sheet sources of meltwater on the basis of estimates of peak GMSL value (e.g., Naish and Wilson, 2009; Raymo et al, 2011; Miller et al., 2012; Grant et al., 2019); for example, many studies attribute peak GMSL of up to approximately +10 m relative to present day to the combined melt from the Greenland and West Antarctic Ice Sheets and any residual GMSL value (i.e., > 10 m above present sea level) to meltwater from the East Antarctic Ice

Sheet. More recent studies have included North American and Eurasian ice cover in the sea level budget (Berends et al., 2019; LeBlanc et al., 2021).

The persistent disagreement among the various mid-Pliocene sea-level reconstructions may stem from limitations of the proxy records that they are derived from, or corrections applied to these proxies. Although $\delta^{18}O$ records

accurately reflect glacial time scales (Lisiecki and Raymo, 2005a; Zachos et al., 2001), numerous complexities introduce errors in the mapping of these records to GMSL (Mix, 1987; Clarke and Marshall, 2002; Waelbroeck et al., 2002; Siddall et al., 2008; Winnick and Caves, 2015). Coupled climate-ice-sea level models rely on accurate proxy measurements and are sensitive to uncertainties in a wide range of model parameters as well as climate forcings (e.g., Berends et al., 2019). Furthermore, an inference of local relative sea level (RSL) based on a

geomorphic or stratigraphic indicators of paleo-sea level is potentially contaminated by three geophysical

processes—tectonics, dynamic topography, and glacial isostatic adjustment (GIA; Raymo et al., 2011; Rowley et al., 2013; Austermann et al., 2017; Richards et al., 2023). Because each process introduces significant geographic variability to sea-level change (i.e., major regional departures from GMSL), any GMSL inference from compilations of geological data are subject to uncertainty and/or error in these geophysical corrections.


In this article, we explore in detail possible geometries of MPWP sea-level change arising from the rotational, gravitational, and deformational effects of the GIA process for a wide range of ice sheet melt zones, including North America, Greenland, Eurasia, West Antarctica, and the Wilkes Basin, Aurora Basin, and Prydz Bay Embayment in East Antarctica. Our focus is on the geometry of sea-level change spanning from the Marine Isotope Stage (MIS)

M2 glacial maximum at 3.295 Ma to the MIS KM3 interglacial at 3.155 Ma, which represent times of peak sea level low and high stand, respectively. These modeling experiments complement the common focus of constraining peak sea level during the KM3 interglacial. We first describe the numerical methods adopted in the study, and the ice history and Earth models that enable sea level predictions. Next, our procedure for normalizing predictions of sea-level change requires a precise definition of GMSL change, and we discuss the definition that we adopted based on

Pan et al. (2022) which, although framed for interglacials, has relevance to the discussion of Pliocene sea-level change. Finally, we present and compare normalized maps of sea-level change for the individual melt zones listed above and discuss the biases in estimates of Pliocene GMSL change that may be introduced by neglecting the geographic variability inherent to these maps.

**2 Methods**

**2.1 Sea Level Model**

Our predictions are based on a generalized form of the sea-level equation (Mitrovica and Milne, 2003; Kendall et al., 2005) that accounts for time-varying shoreline migration and perturbations in Earth's rotation (Mitrovica et al., 2005). We assume a spherically symmetric, Maxwell viscoelastic Earth (Peltier, 1974) and adopt the pseudo-

spectral algorithm described by Kendall et al. (2005) with a truncation at spherical harmonic degree and order 256. The elastic structure of the Earth model is taken from the seismic model PREM (Dziewonski and Anderson, 1981) and, in our primary calculations, the viscosity structure is comprised of a 96-km–thick elastic lithosphere and uniform upper and lower mantle viscosity of $5\times10^{20}$ Pa s and $5\times10^{21}$ Pa s, respectively (henceforth, the 'reference' model). This primary viscoelastic structure is within the range of models inferred from studies of GIA datasets

(Mitrovica and Forte, 2004; Lambeck et al., 2014). However, we also perform an analysis that explores the sensitivity of the normalized sea level predictions to plausible variations in the viscosity model. These additional 27 models are combinations of the following lithospheric thicknesses (72, 96, and 125 km) and upper (0.3, 0.5, and 0.8 Pa s) and lower mantle viscosities (5, 10, and 30 Pa s).

Definitions of how GMSL changes through a deglaciation (or a glaciation) are complicated by contemporaneous changes in ocean area (i.e., shoreline migration) due to local onlap or offlap of water and the advance or retreat of grounded, marine based ice sheets. Figure 1 is a schematic of the primary definition adopted in this study. The

figure shows a cross section through a region with a grounded, marine-based ice sheet that retreats, leading to perturbations in the elevations of the solid Earth and the equipotential that defines the sea surface. We followed Pan
et al. (2022) in defining GMSL change from MIS M2 to KM3 as the mean change in the volume of the ocean outside the grounding line of the ice sheet prior to the melt event (i.e., to the left of the vertical dashed line marked GL on Fig. 1a) divided by the average of ocean area at the beginning and end of the time period of interest (Figs. 1a and b, respectively). (We note that in the simulations we discuss below, the ocean areas at MIS M2 and MIS KM3 differ by less than ~1%, and so choosing to divide by the ocean area at either time instead of taking the average
would have a negligible impact on the normalization procedure.) This definition of global mean sea-level change, henceforth $GMSL_P$, reflects our focus on sea-level changes outside marine-based sectors since it accounts for meltwater sequestered in marine regions exposed by the retreat of grounded ice and the flux out of these areas due to post-glacial rebound. We normalize predictions of the sea-level change from MIS M2 to KM3 by dividing each prediction by the $GMSL_P$ value associated with the GIA simulation. In the Discussion section we consider other
possible definitions for GMSL change.

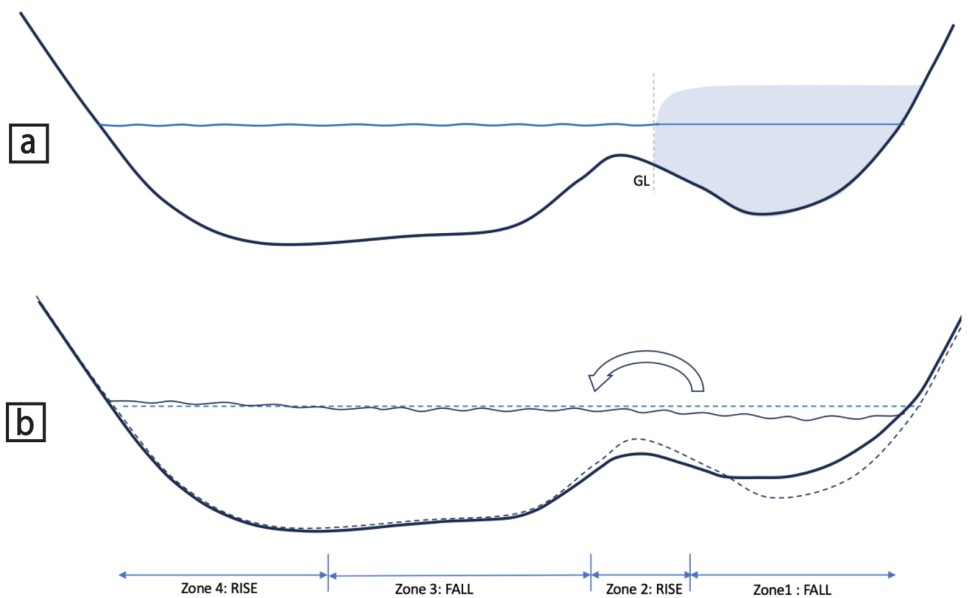

**Figure 1. Sea-level change in response to the melting of a grounded ice sheet.** Sea surface equipotential (blue) and solid surface (black) before (a) and after (b) the melt event. Labeling at bottom of (a) denotes the ice sheet grounding line (GL), and (b) indicates locations of sea-level rise (an increase in the elevation of sea surface equipotential relative to the solid
surface) or fall (elevation of sea surface equipotential relative to solid surface decreases). Zones 1-4 are referred to in the text. The arrow at the top of frame (b) indicates the flow of water into the open ocean driven by the post-glacial uplift of marine sectors previously covered by grounded ice.


| Ice History | GMSL$_P$ (m) | GMSL$_S$ (m) | Enewetak Atoll | | Whanganui Basin | | Virginia | |
|---|---|---|---|---|---|---|---|---|
| | | | LSL (m) | Normalized | LSL (m) | Normalized | LSL (m) | Normalized |
| North America | 32.95 | 34.14 | 34.28 | 1.04 | 29.42 | 0.89 | 20.51 | 0.62 |
| Greenland | 6.66 | 6.70 | 7.00 | 1.05 | 5.75 | 0.86 | 5.60 | 0.84 |
| Eurasia | 4.05 | 4.70 | 4.30 | 1.06 | 3.57 | 0.88 | 3.96 | 0.98 |
| East Antarctica | 11.16 | 12.64 | 11.96 | 1.07 | 9.66 | 0.87 | 10.72 | 0.96 |
| West Antarctica | 2.92 | 4.07 | 3.15 | 1.08 | 2.70 | 0.92 | 3.04 | 1.04 |
| Aurora Basin | 7.04 | 7.37 | 7.44 | 1.06 | 5.93 | 0.84 | 6.45 | 0.92 |
| Wilkes Basin | 5.27 | 5.46 | 5.55 | 1.05 | 4.11 | 0.78 | 4.87 | 0.92 |
| Prydz Bay | 1.89 | 2.21 | 1.97 | 1.04 | 1.72 | 0.91 | 1.76 | 0.93 |

**Table 1. Computed GMSL changes across the ~100 kyr time period extending from MIS M2 to MIS KM3 for eight regional ice histories.** First two columns: GMSL$_P$ (calculated using the reference earth model) and GMSL$_S$. The two definitions of GMSL are defined in the text. Last three columns: predicted LSL changes (in meters) and normalized sea-level change at three sites (Enewetak Atoll, Whanganui Basin and Virginia).


### 2.2 Ice Sheet Model

To explore the GMSL$_P$ value in response to the collapse of an individual Pliocene ice sheet we separately modeled ice sheet variability across eight different regions during the MPWP: Eurasia (EIS), Greenland (GrIS), North America (NAIS), West Antarctica (WAIS), East Antarctica (EAIS) as well as three distinct zones within East

Antarctica, including the Aurora and Wilkes Basins and Prydz Bay. Before any computation was performed, we began by establishing the maximum ice cover of individual ice sheets (GMSL for each is listed in Table 1). The maximum ice volume for each ice sheet occurs at MIS M2 ($\delta^{18}$O value of 3.74‰ in Fig. 2a), whereas the minimum occurs at MIS MG7 (peak-interglacial sea level during our modeled time period; $\delta^{18}$O value of 2.89‰ in Fig. 2a).

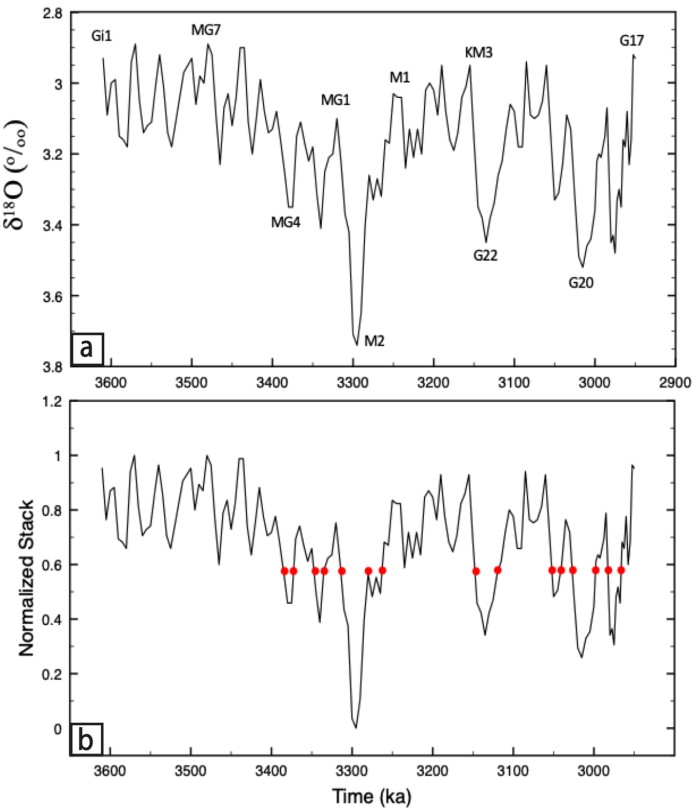

**Figure 2. Time series used in model simulations.** (a) LR04 (Lisiecki and Raymo, 2005) $\delta^{18}$O isotope stack extending from 3610 ka to 2950 ka with labeled Marine Isotope Stage names. (b) Normalized version of the time series in (a), constructed as described in the main text. All points on the time series with the same normalized value have an identical ice geometry (e.g., red points represent those times with a normalized value of 0.6 with precisely the same modeled ice geometry).

Next we adopted a series of Pliocene ice geometries taken from the hybrid ice sheet-climate model results of Berends et al. (2019). These included snapshots of EIS, GrIS, NAIS and Antarctica at MIS M2 and KM3 (Fig. 3), as well as ~25 snapshots at several intervening sea level equivalent ice volumes. Where the maximum M2 SLE ice volume was greater than the Berends et al. (2019) output (e.g., ~34 m SLE from NAIS), additional snapshots with larger ice volumes were supplemented from Berends et al. (2018; e.g., Last Glacial Maximum). The Antarctica ice geometries were first split along the Transantarctic Mountains to produce separate EAIS and WAIS geometries. The EAIS geometries were further broken down by underlying topography to delineate the Aurora Basin, Wilkes Basin and Prydz Bay sub-regions. Additionally, at the MIS MG7 sea-level highstand all ice sheets, except for EAIS, are modeled as entirely deglaciated. For EAIS, whose peak-interglacial melting only involved the marine-based portion, the land-based EAIS (~47 m SLE) always remained during model interglacials (geometry based on the 'PRISM' ice-sheet configuration from Dowsett et al., 2010).

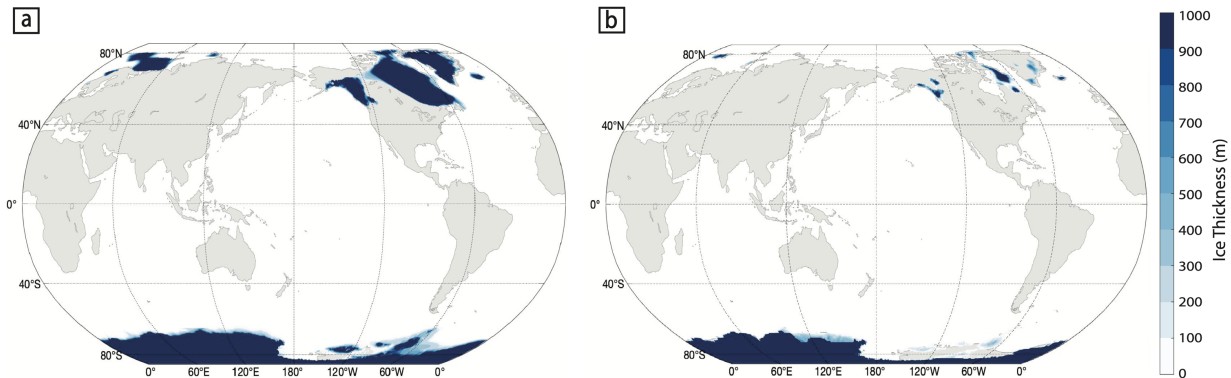

**Figure 3. Modeled Pliocene ice cover during (a) MIS M2 and (b) MIS KM3.** Geometries are based on the hybrid ice sheet-climate model outputs of Berends et al. (2019) as described in the text. Note that in this study each region was modeled separately, but for brevity the ice sheet extents were combined in this figure.

We used the Lisiecki and Raymo (2005) benthic oxygen isotope stack (Fig. 2a) to model the time variation of ice volumes from ~300 kyr prior to MIS M2 (i.e., ~3.6 Ma) to ~200 kyr after MIS KM3 (2.95 Ma). (Ice-volume changes prior to this period would not impact predictions of sea-level change between MIS M2 and KM3.) Specifically, we normalized the magnitude of isotopic variation across this interval to a scale of 0.0–1.0 by subtracting the most depleted $\delta^{18}O$ value (2.89‰ at MIS MG7) from the interval between ~3.6 and 2.9 Ma, then dividing the result by the maximum residual $\delta^{18}O$ value corresponding with the MIS M2 glaciation (3.74-2.89 = 0.85‰), and, finally, subtracting the resulting value from 1.0. This normalized time series is shown in Figure 2b. The SLE ice volumes intermediate between the maximum (MIS M2) and minimum (MIS MG7) glacial conditions in Figure 3 are assumed to scale linearly with the normalized $\delta^{18}O$ time-series and ice geometries are smoothly interpolated across time steps of 1 kyr to accomplish this variation. The construction is performed under the additional constraint that the ice geometry is always the same for the same normalized $\delta^{18}O$ value (e.g., the model ice geometries are identical at each of the times indicated by the red dots on Fig. 2b).

Finally, the global maps of sea-level change calculated for each ice melt scenario are normalized by the GMSL$_P$ value associated with that scenario (Table 1). Since the sea level predictions are quasi-linearly related to the net ice mass flux, this normalization procedure yields maps that are - outside the immediate vicinity of the melt zone - relatively insensitive to the GMSL change, or equivalently the total ice mass flux, of the scenario. We demonstrate this insensitivity in the results below. The linearity also allows one to combine, with suitable weighting, the maps for individual melt zones, to assess the connection between LSL change at any site and total GMSL$_P$ for any scenario of interest. This generality is an important point to emphasize because we make no assertion regarding the validity of the total melt volumes in each of the eight scenarios listed in Table 1, and our main conclusions regarding biases in the mapping between local and global sea level are insensitive to these melt volumes.

With respect to the normalized $\delta^{18}O$ time series utilized in this study, there are uncertainties in the LR04 stack derived frequency and amplitude of 3.3-3 Ma glacial-interglacial cycles. The stack was put together from 57 different benthic $\delta^{18}O_{carb}$ and Mg/Ca ratios (Lisiecki and Raymo, 2005), and is complicated by uncertainties in fossil

species and proxy specific calibrations, alteration due to diagenesis, and changes in seawater chemistry (Raymo et
al., 2017). Additionally, studies of ice-berg rafted debris from areas proximal to the EAIS suggest that, unlike $\delta^{18}O$
records over the 3.3-3 Ma time period, glacial-interglacial cycles were not paced by obliquity (40 kyr) but instead
(23 kyr) precession (Patterson et al., 2014). Therefore, to accommodate these uncertainties we performed sensitivity
analyses in which we shortened the time duration between MIS M2 and KM3 from 140 kyr to 120 kyr or randomly
perturbed the magnitude of the smaller sea-level oscillations between the two marine isotope stages (Fig. 2).

**3 Results**

Figure 4 shows maps of sea-level change computed for the eight different regional ice histories normalized by the
GMSL$_P$ value associated with each ice history (Table 1). These plots can be interpreted as 'viscoelastic' fingerprints
that include both the viscous and elastic effects through the MIS M2 – KM3 period. (The term 'viscoelastic'
fingerprint is used to distinguish the maps from commonly published 'elastic' fingerprints which are computed for
melt events sufficiently rapid that viscous effects can be ignored).

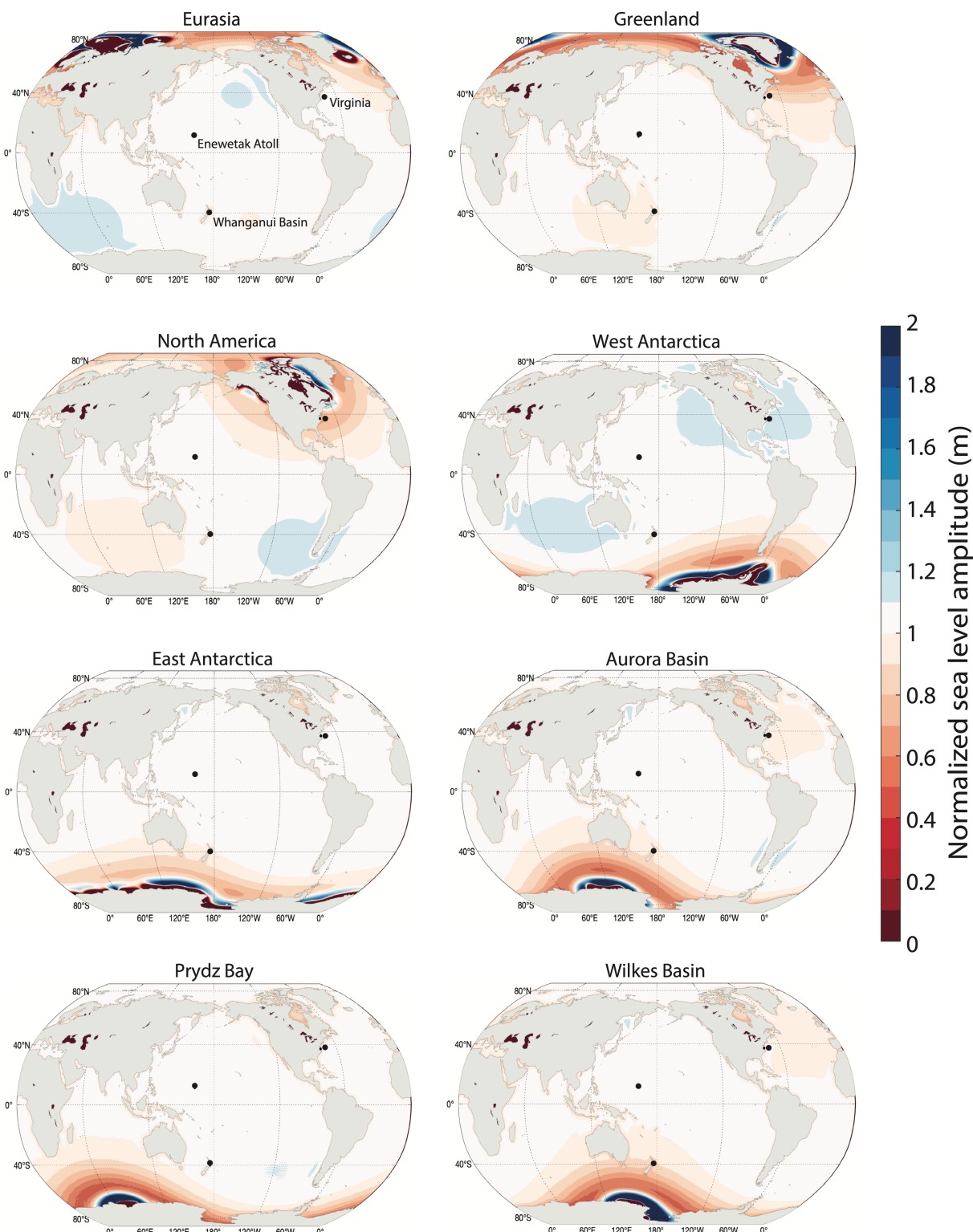

**Figure 4. Predicted sea-level change from MIS M2 to MIS KM3 for eight different regional ice histories (as labeled, above each frame).** Predictions are based on the reference viscoelastic Earth model described in the text and, to facilitate comparison, are normalized by the $GMSL_P$ value associated with each simulation (Table 1). The three black dots on the figure show the location of continental shelf/upper slope sites discussed in the text.

The normalized maps in Fig. 4 show similar structures in relation to the zones of ice mass flux. In the area once covered by ice, a sea-level fall of high magnitude (off the scale of the plot) is predicted and as one considers sites progressively further from this region, zones of sea-level rise (blue, which also reaches amplitudes off scale) and fall (light to dark red) are predicted. Superimposed on these trends is a so-called "quadrantal" (spherical harmonic degree two, order one) sea level pattern due to true polar wander (TPW; Milne and Mitrovica, 1996). TPW contributes a sea-level fall in the quadrant encompassing ice melt and in the anti-polar quadrant, and a sea-level rise in the remaining two quadrants. As an example, melting over Laurentia contributes a TPW-induced sea-level fall over North America and the southern Indian Ocean and a sea-level rise centered over southern South America and southeast Asia.

Putting aside the TPW signal, the origin of the complex trends in the predicted sea-level change as one moves from the near to far field of an ice sheet (Fig. 4), which are characterized by several changes in sign, is captured in the schematic of Figure 1. The total change in sea level can be understood as having two contributions. First, a reduction in the ice mass from MIS M2 to KM3 leads to a migration of water from the near to far field as the gravitational pull of the ice sheet weakens. This leads to a long wavelength tilting of the sea surface up-toward-the-far field on Figure 1b (blue wavy line). Second, superimposed on this gravitational signal, is viscous deformation comprised of post-glacial rebound in the zone of ice retreat (zone labeled 1), subsidence of a peripheral bulge (zone 2), and relatively minor crustal subsidence due to ocean loading (zone 3). In zone 1, post-glacial rebound and the loss of gravitational pull on the ocean combine constructively to produce a sea-level fall with a peak amplitude more than 10 times greater than the GMSL rise of the ice history (red, largely covered by the continental mask used on the figures). In zone 2, peripheral subsidence is of greater magnitude than the water migration away from the near field and the result is a predicted sea-level rise in the maps of Figure 4 (blue contours). In zone 3, the opposite happens; the long wavelength tilting of the sea-surface (and migration of water) due to the loss of gravitational pull toward the ice sheet once again dominates crustal subsidence and a sea-level fall is predicted (red zone encircling the blue). In zone 3 the predicted sea-level fall also has a contribution from ocean syphoning, the movement of water away from these regions into the accommodation created primarily by the subsiding peripheral bulges (Mitrovica and Milne, 2002). Finally, water migration into zone 4 dominates other effects and sea level rise occurs.

As discussed in the Introduction, the normalization procedure applied in each scenario within Figure 4 should yield maps that are relatively insensitive to changes in the net volume of melt if the geometry of the ice melt is not fundamentally altered. To highlight this issue, Supplementary Figure 1 (Fig. S1) shows a map analogous to the NAIS scenario in Figure 4 with the exception that we adopted a melt model with a $GMSL_P$ value of 7.71 m. Outside of the region in the near vicinity of the mass flux, the two normalized maps show nearly identical structure. Of course, the sensitivity is larger at sites close to the mass flux, as we discuss below. Additionally, the sensitivity analyses with a 120 kyr time duration and smaller sea-level oscillations between MIS M2 and KM3 (Fig. 2) revealed that the normalized sea level maps were negligibly impacted.

The viscoelastic fingerprint maps in Figure 4 exhibit significant departures from GMSL for the period extending from MIS M2 glacial maximum to MIS KM3 glacial minimum. The geographic pattern of these departures is governed by the location of the modeled ice melt and we next turn to the implications of this variability on inferences of the total amplitude of GMSL change that might be inferred from local geological indicators of sea-level change. To broaden our assessment of this issue, we incorporated the 27 additional simulations of variable

lithospheric thickness and mantle viscosity discussed above (see Methods for values). Figure 5 shows, for all eight regional ice histories, the full range of normalized sea level predictions for all 27 earth models at three sites that host MPWP stratigraphic indicators—one in the near field of Northern Hemisphere ice sheets (Virginia), one in the near field of Antarctica (Whanganui Basin), and one in the far field of all ice sheets (Enewetak Atoll). We note that an inference of sea-level change across the MIS M2 to KM3 interval has only been made for Whanganui Basin (Grant

et al., 2019); the additional two locales offer an illustration for how the predicted amplitude of local sea-level change across the modeled MIS M2 to KM3 interval would deviate from GMSL$_P$ for the scenarios considered here. The range of the 27 predictions, each normalized by the GMSL$_P$ value of the scenario, is summarized by a box and whiskers plot (Fig. 5). The black circle within the box and whisker plot refers to the value of sea-level change for the reference earth model and individual ice sheet, while the black line demonstrates the median value for all 27

earth models for an individual ice sheet. The normalization procedure allows us to meaningfully compare the results across these models.

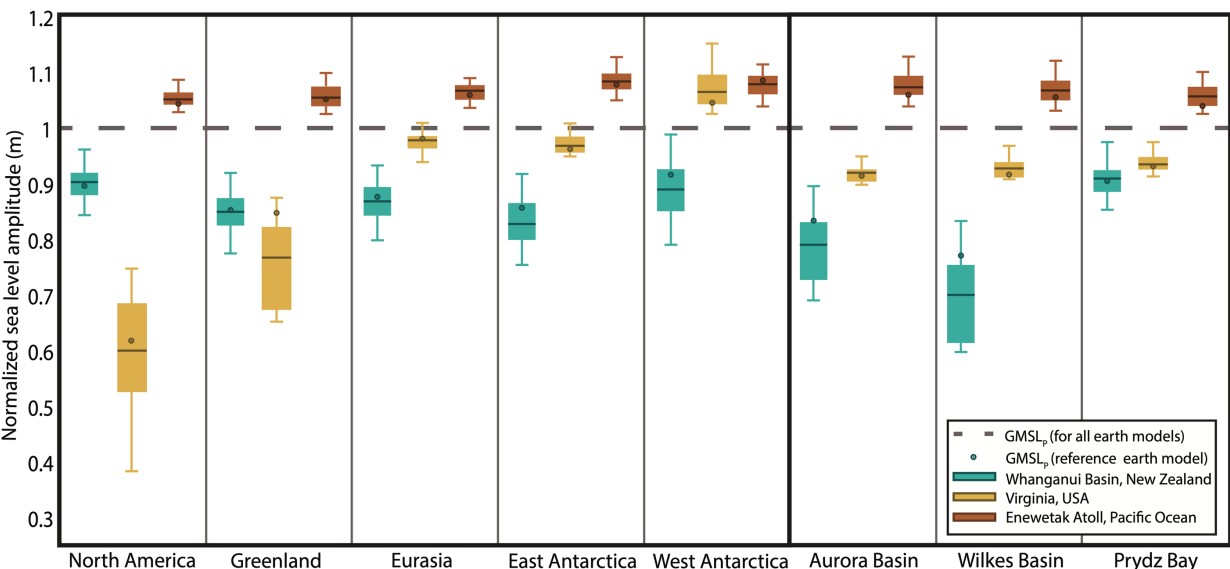

**Figure 5. Predicted MIS M2 to MIS KM3 sea-level changes for three geographic sites (see inset key, and Fig. 4 for**
**locations) based upon a range of melt and viscoelastic Earth models.** The box-and-whisker plots show the range of results generated using the 27 different viscoelastic models (discussed in the text). All predictions are normalized by the global mean change, GMSL$_P$, associated with the specific melt and Earth models. The dashed line denotes the result that would occur if the prediction matched the associated GMSL$_P$ value and, thus, departures from 1.0 represent normalized (fractional) departures from the global mean change in sea level as defined by Pan et al. (2021).

Predictions at Enewetak Atoll, in the very far field of ice mass changes are consistently ~0-15% greater than GMSL$_P$ (Fig. 5). This site is within zone 4 of Figure 1 but the prediction is influenced in some simulations by rotational effects (Fig. 4). The predictions of sea-level change at Whanganui Basin have a larger spread than those

for Enewetak and are consistently below the global mean (GMSL$_P$) for all melt models and for all Earth models. In the case of melting in the northern hemisphere (e.g., EIS, GrIS and NAIS melt models) the departure from GMSL$_P$ is dominated by the sea-level fall associated with rotational effects (Fig. 4). These effects also contribute to the results for southern hemisphere melt models, but in those cases the migration of water away from the zones of melt tends to dominate (Fig. 1; zone 3), particularly in the case of melt from the Aurora and Wilkes Basins (Figs. 4 and 5). In the case of these melt zones, the local prediction at Whanganui Basin varies from ~60-98% of the global mean value. Finally, the predictions at Virginia, on the United States' east coast, show even greater sensitivity to the location of melt. In the case of the simulations involving melt from NAIS or GrIS, the prediction is dominated by the migration of water away from the area of melt (Fig. 1; zone 3) and rotational effects, which lead to a sea-level change substantially lower than GMSL$_P$ (Fig. 5). Rotational effects dominate the departure from the global mean and contribute a sea-level fall for all cases of melt within the East Antarctic and a sea-level rise for melt sourced from West Antarctica. We emphasize that these three sites are chosen as illustrative case studies, and that the maps in Figure 4 can be used to assess the relationship between LSL and GMSL$_P$ for any site and for any of the eight melt scenarios.

As a further illustration of the utility of the maps in Figure 4, Figure 6 plots the maximum discrepancy of computed sea-level change from the total GMSL$_P$ based on the reference Earth model and the following unweighted combinations of ice melt models: GrIS, WAIS, and EAIS (Fig. 6a); and NAIS, EIS, GrIS, WAIS, and marine-based EAIS (Fig. 6b). (One can repeat the same exercise with any weighted combination of the maps in Fig. 4.) The first combination of ice melt sources reflects the view that only the modern-day ice sheets contributed melt from MIS M2 to the KM3 interglacial. The second combination incorporates a contribution from two additional ice sheets (NAIS and EIS) across this time period since recent studies have included NAIS and EIS contributions to the sea level budget (Berends et al., 2019; LeBlanc et al., 2021). The two maps identify geographic regions in which the LSL variation might provide the closest measure of GMSL change from MIS M2 to KM3. For both scenarios, the maximum discrepancy is highest within the near field of the modeled ice mass flux (both scenarios yield discrepancies greater than 20% at, for example, the Antarctic coastline) and lowest in equatorial regions in the far field. In both scenarios, areas in the Indian Ocean extending from Indonesia to Papua New Guinea, South Pacific Ocean from 180-150°W, and some equatorial coastlines are predicted to have experienced a sea-level change from MIS M2 to MIS KM3 within 5% of the global mean value. In contrast, the discrepancy is large (>10%) along the remaining global coastline. Additionally, at Enewetak Atoll and Whanganui Basin, the scenarios yield consistent deviations of up to 15% from GMSL$_P$ (Fig. 6). (Note that for the Whanganui Basin the colored area indicating 15% is partially obscured by the land mask). These discrepancies, and indeed the departure from GMSL$_P$ from any other combination of melt sources at these sites, can be inferred from the individual ice sheet results in Figure 5.

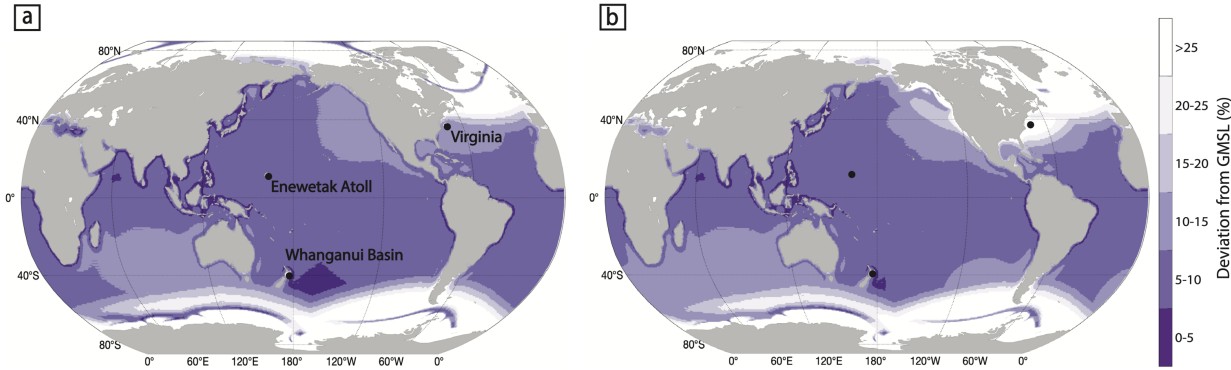

**Figure 6. The maximum percent discrepancy of the reference viscoelastic Earth model predictions of LSL change from GMSL$_P$ across MIS M2 to MIS KM3.** The following scenarios of ice melt locations are commonly forwarded in published literature: (a) Greenland, West Antarctica, and marine-based East Antarctica; and (b) North America, Eurasia, Greenland, West Antarctica, and East Antarctica.

Since Whanganui Basin is the only site with a published estimate of sea-level change across the MIS M2 – KM3 deglaciation (Grant et al., 2019), we further explored the discrepancy between global and LSL at this site. It is clear from Figure 5 that any inference of the LSL change at this site will always be smaller than the GMSL$_P$ value. To highlight possible departures of site-specific observations from GMSL$_P$, Figure 7 includes five example scenarios where combinations of ice sheet melt, in conjunction with the reference earth model, predict a ~15 m amplitude LSL rise across this deglaciation at Whanganui Basin. The five scenarios presented in Figure 7 were chosen to represent one set of commonly accepted sources of Pliocene ice sheet melt (a), a scenario that excludes ice sheet contributions from North America (b) and East Antarctica (c), and two scenarios that includes melt from all ice sheets (d and e). Bar plots (Fig. 7f) provide the GMSL$_P$ value from each ice sheet in a given scenario (a-e), as well as the total. This result emphasizes the systematic difference between LSL change at Whanganui Basin and GMSL. In these scenarios, the 15 m LSL change at Whanganui Basin is consistently ~12% lower than GMSL$_P$ (16.94, 17.07, 16.92, 17.00, and 17.01 m, respectively, in Fig. 7f).

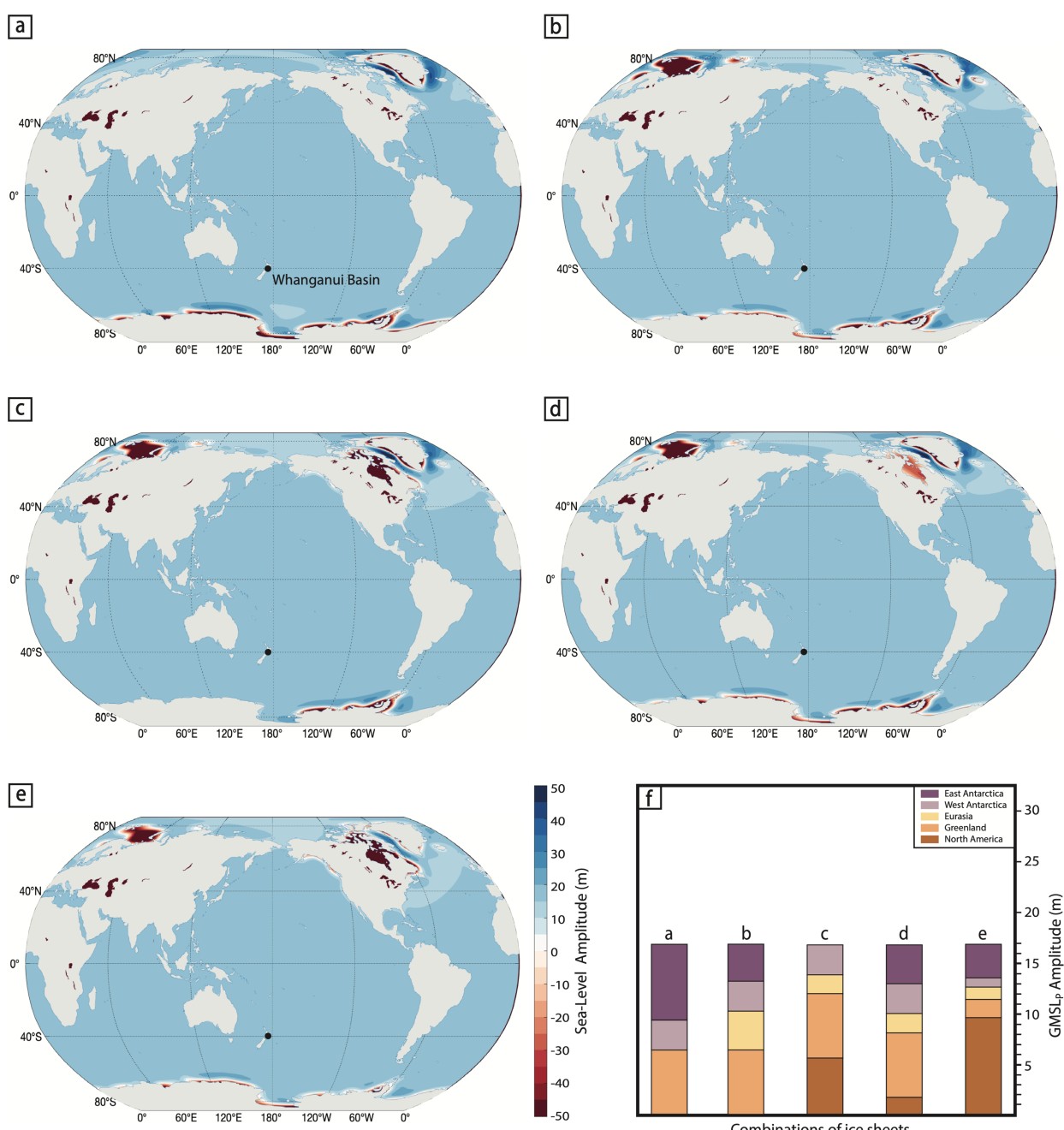

**Figure 7. Catalog of ice-sheet combinations that produce 15 m amplitude LSL change at Whanganui Basin, New Zealand.**
(a-e) Global maps of the total sea-level change from the MIS M2 glacial to the MIS KM3 interglacial for the five scenarios of ice sheet melt. Frame (f) shows the global mean change ($GMSL_P$) associated with each of the 5 melt scenarios and the contribution to this value from each region of melt.

## 4 Further Discussion and Conclusions

Our analysis has highlighted the geographically variable change in sea level associated with a variety of potential meltwater sources to a major MPWP glacial–interglacial cycle. This variability provides a direct measure of the departure of LSL rise from the global mean change anywhere in the global ocean (Fig. 4), including sites that have

contributed to estimates of peak and glacial cycle sea-level change during MPWP. Discussions of this departure require a robust and transparent definition of GMSL change. The definition we have adopted, GMSL$_P$, involves dividing the total meltwater volume that enters the open ocean outside any exposed marine based sectors from MIS M2 to KM3 by the area of the ocean (Fig. 1). The appropriateness of this choice is suggested by the normalized sea-level change maps of Figure 4, which are all characterized by values within a few percent of 1.0 along the equator. (Mean equatorial ocean values are: Eurasia: 0.9870, Greenland: 0.9695, North America: 0.9703, West Antarctica: 1.0150, East Antarctica: 1.0000, Aurora Basin: 0.9807, Prydz Bay: 0.9835, and Wilkes Basin: 0.9717 for the normalized maps in Fig. 4.) That is, at sites furthest afield from the deformational, gravitational and rotational effects of the GIA, the calculated sea-level change reflects the GMSL change.

Other definitions of GMSL change are, of course, possible. Figure S2 extends Figure 1 to include two other possibilities. The first, GMSL$_{IAF}$, involves spreading the ice volume above floatation as defined at the start of MIS M2 over the global ocean. This ignores the flux of water from exposed marine sectors which will be a significant limitation considering the time duration of the MPWP interval we are considering (~140 kyr) in scenarios with considerable ice sheet retreat from such sectors. The second, GMSL$_S$, takes the full volume of meltwater between MIS M2 and KM3 and spreads it over the ocean. As in the case of GMSL$_P$, the area of ocean used in the calculation (i.e., whether or not the marine sector is included) will have ~1% or less effect on GMSL$_S$. One can interpret GMSL$_S$ as a special case of GMSL$_P$ in which any exposed marine based sectors rebound sufficiently in the calculation of GMSL$_P$ that they become subaerial. This will, of course, depend on the volume of the marine accommodation space relative to the total post-glacial uplift of the crust from MIS M2 to KM3. Table 1 also shows the GMSL$_S$ value computed for each ice history described above. The limitation of adopting this definition is most pronounced in the results for West Antarctica, where substantial marine-based regions are exposed across the ice history. The difference in the GMSL calculations (4.07 - 2.92 ~ 1.15 m) largely reflects, in the calculation of GMSL$_P$, the volume of meltwater that remains in these marine-based sectors at MIS KM3 that were exposed by grounded ice retreat from MIS M2 to KM3. If one were to use GMSL$_S$ instead of GMSL$_P$, then the normalized map of the WAIS scenario in Figure 4 would show values of ~0.7 (2.92/4.07) rather than 1.0 near the equator, i.e., the "far field", which suggests that GMSL$_S$ is not an appropriate metric for GMSL change in this case. The metric GMSL$_P$ yields values intermediate to GMSL$_{IAF}$ and GMSL$_S$ and all three definitions of GMSL change will be identical in the case where no grounded, marine-based ice is involved in an ice melt scenario. The latter is close to being the case in the GrIS scenario we have adopted.

The Whanganui Basin hosts well-preserved Pliocene continental shelf stratigraphy (Naish and Wilson, 2009). Assuming the modern wave climate was similar to the Pliocene, Grant et al. (2019) applied a theoretical relationship between modern sediment transport by waves and water depth to temporal variation in grain in Pliocene core/outcrop samples. This method applied a two-dimensional backstripping method to correct for the effects of tectonic subsidence and sediment compaction to estimate the amplitude of MIS M2 to MIS KM3 LSL change of 13 ± 5 m. Grant et al. (2019) noted that, while their analysis strictly provided a measure of *local* RSL change, their

modeling of GIA indicated that the reconstruction also served as a good approximation of GMSL and, thus, ice-volume fluctuation. The results of Figures 4–7 indicate that this local measurement will be lower than the associated $GMSL_P$ value by an average of ~15%.

Beyond a robust estimate of GMSL change across the MIS M2 to KM3 deglaciation, a further goal of MPWP paleo-

sea level studies is to constrain the sources of ice mass flux and their relative contributions. For a given site, the greater (smaller) the spread of the box-and-whisker predictions across the various melt scenarios (Fig. 5), the greater (lower) the ability of that observation, when viewed in combination with other observations, to constrain the contributors to the sea-level change from the MIS M2 glacial to the MIS KM3 interglacial. As an example, an accurate observation at Enewetak Atoll would provide a powerful constraint on $GMSL_P$ because all melt zones

provide a consistent scale factor between LSL change and $GMSL_P$. Yet that consistency indicates, conversely, that this datum provides no discriminatory information on the melt source(s). Combining an observation at Enewetak atoll with one at Virginia, and/or Whanganui Basin might yield both a strong constraint on $GMSL_P$ and narrow the possible sources of melt. Further exploration of the results of Figure 4 will provide other potential sites that can contribute to establishing such constraints in future work.


**Code and Data Availability**

Data for the ice and sea level models, as well as code used to produce figures will be available at https://github.com/meghan-king/plioceneSeaLevel upon publication.

**Author Contribution**

MEK, JRC and JXM designed the study and MEK performed all the simulations. MEK prepared the manuscript and figures with contributions from JRC and JXM.

**Competing Interests**

The authors declare no competing interests with respect to the results of this paper.

**Acknowledgements**

We thank C.J. Berends for the MPWP ice snapshots. We also thank T.R. Naish and an anonymous reviewer for their constructive comments which improved this manuscript.


**Financial Support**

This research was made possible by U.S. National Science Foundation award 2046244, a Geological Society of America (GSA) graduate student research grant, and the Oregon State University George and Danielle Sharp Fellowship.

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

## Supplementary Material

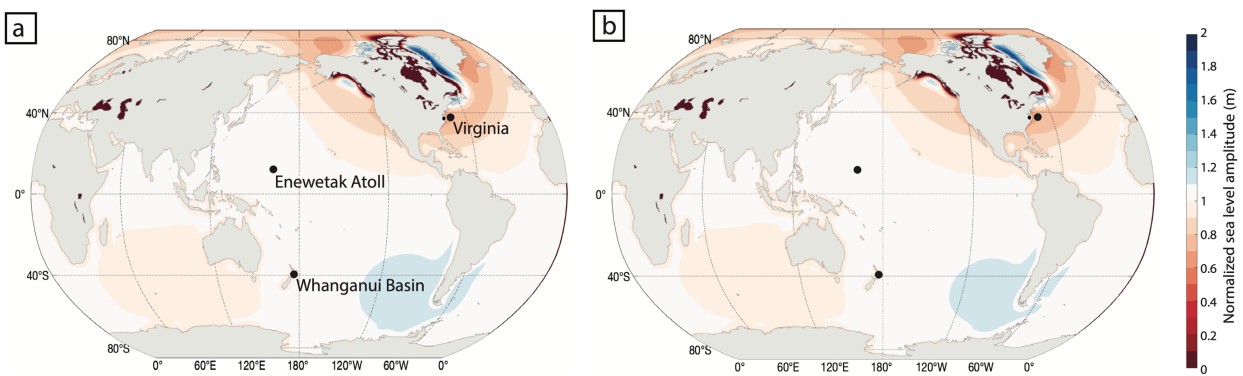

**Figure S1.** Comparison of normalized sea level maps for MIS M2 to KM3 NAIS collapse where GMSL$_P$ is (a) 32.95 m and (b) 7.71 m. Predictions are based on the reference viscoelastic Earth model described in the text, and the three black dots on the figure show the location of continental shelf/upper slope sites discussed in the text.


**Figure S2. Alternative definitions for global mean sea-level (GMSL) change.** Left – GMSL$_{IAF}$ involves melting of the grounded, marine based ice sheet and spreading the meltwater over the ocean to fill accommodation space under the assumption that the solid Earth and gravity field remains unperturbed. Right – GMSL$_S$ is similar to GMSL$_{IAF}$ except that the entire volume of meltwater is spread over the global ocean. The assumption inherent to this definition is that the exposed marine sector does not accommodate meltwater. GL = grounding line.