# Peer review of "The geometry of sea-level change across a mid-Pliocene glacial cycle"

_EGUsphere, 2024_

## Referee Comment (RC1)

This paper is a nice model approach using state of the art GIA model to reconstruct GRD effect son global sea-level change during the time interval spanning the MIS M2 glacial to MIS KM3 interglacial as defined in the L&R05 d180 stack. Even though I have outlined below, why I beleive the methodology is flawed, I would like to encourage the authors to consider trying a diffrent range of ice sheet histories that might better reconcile with the far-field geological record of sea-level chnage. Its always difficult using a GIA model to evaluate a sea-level record when the ice sheet history is ambiguous.

I will declare up front that I am Tim Naish, and have been closely associated with the development of Whanganui Basin, NZ sea-level records. I also saw Meghan King present this paper at Fall AGU on 2022, where I discussed it briefly with her afterwards. I remian supportive of her work. I dont feel conflicted, but will leave that up to teh eds to decide.

It might help also if I mention the motivation behind the 2019 Grant et al study published in Nature. We were well-aware that the L&R05 d18O stack was of lower qulaity between 3.3-3 Ma due to low number of records and poor resolution of some of the records. The shallow marine glacial-interglacial sedimentary cycles in Whanganui are well dated in this interval as both Kaena and Mammoth paleomagnetic subchrons could be indentified as well as radiometrically-dated tephra. We argued for 2kyr sample resolution with an accuracy of +/- 4-5kyr at pmag transitions. On this basis we built an independently dated sea-level record and showed that it was largely in phase with Antarctic summer insolation. Given geological evidence precluding a large NH ice sheet until 2.7Ma (*Haug et al., 2005; Jansen et al., 2000; Brigham-Grette et al., 2013; Berends et al., 2019, Eldrett et al., 2007; Thiede et al., 2011; Tripati & Darby, 2018*). we also argued most of the meltwater was of Antarctic origin, and this informed the GIA modeling to we used to test how close Whanganui RSL was to ESL.

My comments follow.

Line 100. Definition of global mean sea-level needs to be registered to the centre of the Earth, otherwise it is eustatic sea-level. Im OK with Pan et al 2022 method for estimating change in eustatic sea-level ESL (but shouldnt use GMSL unless you can register it to presnet day sea-level

Line 145. The Brerends et al. ice sheet histories puts more ice on northern hemishphere continents during M2 than geological evidence implies (see above refs).

Line 160. Uses the assumptiion that L&R05 is valid time variation in global ice volume for this time interval. Other studies have raised concerns about the frequency and amplitude of the stack between 3.3-3 Ma, where d18O records are low in number and resolution, leading to potential artifacts through the stacking process (Patterson et al., 2014, Nature Geo; Grant & Naish, 2021, Pages). Note that the highest resolution d18O record during this time interval is dominated be prcession (ODP 846-849), as it should be due to a node in obliquity in the orbital solution. Note also the Grant et al 2019 sealevel record which is independent of the L&R05 stack correlates strongly with high southern latitude insolation dominated by precession.

Note also that the proxy ice sheet histories for this time inetrval are not well constrained by d18O or the Breneds modelling, or the ice berg rafted debris records from Antarctica and the Arctic (this should be brought into thediscussion). Certianly high latitude northern hemsiphere IBRD records (N Pacific, Norwegian Sea, E Greenland) show no contiental scale NH ice sheet margins (as implied by Berends et al), with the exception of Greenland. This is why there should be caution taken rather just than accepting Berends et al., as a "series of Pliocene-realistic ice geometries" (line 144).

Line 250. The age model for Enewetak Atoll has always been very uncertain. Table 3 of Wardlaw and Quinn paper is very hard to understand. The "mid-Pliocene" 3.6-3.5 Ma range of RSL change is -33m to +25m (extreme). This is not an equivalent interval to the M2-KM3.

 If I understand correctly this paper is modelling for the interval M2 to KM3 which is 3.3-3.15 Ma. In Grant et al *PlioSeaNZ* record this represents a change from +3 to +23m ESL (an overall increase of 20m over 8 eccentricity-modulated  precision cycles).  However, it should be  noted that the PlioSeaNZ record floats, and while it constrains the amplitude and timing of ESL  during G-I variability, it is only registered to present day sea-level through an assumption (see final paragraph). This paper would be greatly improved by the inclusion of a table showing the data used for amplitude and age of sea-level  in the proxy records (e.g. Whanganui Basin,  Enewetak Atoll).

Line 290, The authors claim a 20% underestimation in GMSL  (should be eustatic sea-level as stated above) from the Whanganui Basin, PlioSeaNZ record, which is unsurprisingly consistent with their ice sheet histories. However they should note,  that Grant et al. 2019 used a series of quite different hypothetical  ice sheet histories in their GIA modelling, which were based on best estimates from geological reconstructions, whereby +20 m of ESL was released under 4 scenarios **1**), 20 m ESL released from AIS only. **2**, AIS and GIS synchronously release 15 m and 5 m ESL, respectively **3**, AIS releases 25 m ESL while GIS accumulates 5 m ESL (that is, in antiphase). **4**, AIS and NHIS synchronously release 10 m ESL. In all cases the Whanganui Basin record lay geographically on the eustatic.

Any GIA model with an ice sheet history releasing 70% of the melt water from the northern hemisphere (such as in this paper) will overestimate the Whanganui record. Grant et al 2019 would have done the same had they used the ice sheet history used in Table 1 of this paper, which requires a total of 70m SLE ice is being melted between M2 and KM3. This ice sheet history is not supported by the published geological constraints. For example the total from Antarctic sector loss is greater than the total ice volume currently held in all the marine-based sectors and would require M2 glacial to be larger than present day ice volume, as well as having 45m SLE ice on the northern hemisphere continents.

Notwithstanding all of the above, paper is a very nice piece of work, and should be published, if the authors can consider the following. That the...

1. ice sheet history used based on Berends et al is a modelled outcome and not supported by the weight geological data. Therefore, the result of this modelling exercise is a function of the ice sheet history used rather than a true test of the far far-field sea-level proxies.
2. The Enewetak sea-level reconstruction is for a different part of the Pliocene (not M2 to KM3), it is like comparing apples with oranges.

---

## Author Comment (AC1)

Our manuscript titled "The geometry of sea-level change across a mid-Pliocene glacial cycle" received two largely positive reviews that listed several suggestions for improvement. In the material below, we respond to the first comment. The reviewer comments appear in blue, and our responses are in black, with text quoted from the manuscript indented.

**Reviewer #1 (Dr. Naish):**

This paper is a nice model approach using state of the art GIA model to reconstruct GRD effects on global sea-level change during the time interval spanning the MIS M2 glacial to MIS KM3 interglacial as defined in the L&R05 d180 stack. Even though I have outlined below, why I believe the methodology is flawed, I would like to encourage the authors to consider trying a different range of ice sheet histories that might better reconcile with the far-field geological record of sea-level chnage. Its always difficult using a GIA model to evaluate a sea-level record when the ice sheet history is ambiguous.

I will declare up front that I am Tim Naish, and have been closely associated with the development of Whanganui Basin, NZ sea-level records. I also saw Meghan King present this paper at Fall AGU on 2022, where I discussed it briefly with her afterwards. I remian supportive of her work. I dont feel conflicted, but will leave that up to teh eds to decide.

It might help also if I mention the motivation behind the 2019 Grant et al study published in Nature. We were well-aware that the L&R05 d18O stack was of lower qulaity between 3.3-3 Ma due to low number of records and poor resolution of some of the records. The shallow marine glacial-interglacial sedimentary cycles in Whanganui are well dated in this interval as both Kaena and Mammoth paleomagnetic subchrons could be indentified as well as radiometrically-dated tephra. We argued for 2kyr sample resolution with an accuracy of +/- 4-5kyr at pmag transitions. On this basis we built an independently dated sea-level record and showed that it was largely in phase with Antarctic summer insolation. Given geological evidence precluding a large NH ice sheet until 2.7Ma (Haug et al., 2005; Jansen et al., 2000; Brigham-Grette et al., 2013; Berends et al., 2019, Eldrett et al., 2007; Thiede et al., 2011; Tripati & Darby, 2018). we also argued most of the meltwater was of Antarctic origin, and this informed the GIA modeling to we used to test how close Whanganui RSL was to ESL.

My comments follow.

We thank Dr. Naish for his review and constructive comments.

Line 100. Definition of global mean sea-level needs to be registered to the centre of the Earth, otherwise it is eustatic sea-level. Im OK with Pan et al 2022 method for estimating change in eustatic sea-level ESL (but shouldnt use GMSL unless you can register it to present day sea-level).

In studies of paleo sea level global mean sea level changes are not referenced to the center of the Earth. Any change in the elevation of a geological marker reflects *relative* sea level change – that

is, the change in the sea surface height relative to any crustal elevation change. In modern sea level studies, tide gauge records also reflect the change in the distance between the sea surface and crust, while satellite records measure "*absolute* sea level" changes, i.e., changes in the sea surface height relative to some reference frame (e.g., the center of the Earth).

In the geological literature of long term sea level change there has been an argument that only the change in the sea surface height, or absolute sea level, is important (e.g., http://stratigrafia.org/sequence/accommodation.html) but this doesn't hold if there is any departure at all from hydrostatic (not just isostatic) equilibrium. In an ice age calculation, there is always such a departure because of the adoption of an elastic lithosphere and unrelaxed viscous stresses.

Perhaps the issue arises from the fact that our text does not always make clear that we are speaking of global mean sea level *changes*. To address this, we will revise the text to ensure that all mentions of global mean sea level are connected to the word "change".

Finally, we note that the terms "eustatic" and "eustasy" are being avoided because of the ambiguity associated with their definition (Gregory, et al. "Concepts and terminology for sea level: Mean, variability and change, both local and global." *Surveys in Geophysics, 40,* 1251-1289, 2019).

Line 145. The Brerends et al. ice sheet histories puts more ice on northern hemisphere continents during M2 than geological evidence implies (see above refs).

Yes, the Berends et al. (2019) ice sheet histories certainly suggest significantly more Northern Hemisphere (i.e., NAIS and EIS) ice than other studies. The reviewer's comment does, however, point to another way in which our manuscript can be improved. The normalization procedure we apply to the raw sea level calculation to yield Fig. 4 allows us in principle to scale these figures by any global mean sea level scenario since the sea level calculations are (quasi) linearly related to the total mass flux. As an illustration of this, the figure below compares the normalized sea level map of the NAIS scenario from Fig. 4 of the manuscript (GMSL$_P$ = 33.12 m) with the equivalent map for an alternate NAIS scenario characterized by GMSL$_P$ = 9.2 m. The two maps are nearly identical in areas away from the ancient ice cover. So, in this sense, even if the 33 m scenario is considered unrealistic, the normalized sea level change holds for any other scenario, at least for sites away from the ice cover.

[Figure]

Figure S1. Comparison of normalized sea level maps for MIS M2-KM3 NAIS collapse where GMSL$_P$ is (a) 33.12 m and (b) 9.2 m.

We will include this new material in the main text of the revised manuscript to make the point that the normalized sea level map for the NAIS scenario is not very sensitive to a plausible range in the global mean sea level change assumed for the scenario.

Line 160. Uses the assumption that L&R05 is valid time variation in global ice volume for this time interval. Other studies have raised concerns about the frequency and amplitude of the stack between 3.3-3 Ma, where d18O records are low in number and resolution, leading to potential artefacts through the stacking process (Patterson et al., 2014, Nature Geo; Grant & Naish, 2021, Pages). Note that the highest resolution d18O record during this time interval is dominated be precession (ODP 846-849), as it should be due to a node in obliquity in the orbital solution. Note also the Grant et al 2019 sea-level record which is independent of the L&R05 stack correlates strongly with high southern latitude insolation dominated by precession.

Whereas the last comment dealt with the net magnitude of the global mean sea-level change between MIS M2 to KM3, this comment is focused on the details of the sea-level oscillations between these two times. These details may be subject to error, as pointed out by the reviewer. Calculations performed as we prepared our original manuscript indicate that the multiple small oscillations have only a minor impact on the normalized sea level maps of Fig. 4. The primary control on these maps is the time duration between MIS M2 to KM3, which we have modeled as 140 kyr. In the revised manuscript we will make this point by including a series of supplementary calculations in which the magnitude of the multiple smaller sea level oscillations between MIS M2 and KM3 - and the total time difference between these stages – are varied to explore the impact of the associated normalized sea level maps.

Note also that the proxy ice sheet histories for this time interval are not well constrained by d18O or the Berends modelling, or the ice berg rafted debris records from Antarctica and the Arctic (this should be brought into the discussion). Certainly high latitude northern hemisphere IBRD records (N Pacific, Norwegian Sea, E Greenland) show no continental scale NH ice sheet margins (as implied by Berends et al), with the exception of Greenland. This is why there should be caution taken rather just than accepting Berends et al., as a "series of Pliocene-realistic ice geometries" (line 144).

The new material (and figure) discussed above will emphasize that uncertainties associated with the size of Pliocene NH ice sheets will not map into large uncertainties in the the normalized sea level-change maps. As part of this revised text, we will include a discussion of proxy records that argue against the existence of a large NH ice sheet.

Nevertheless, we note that since we model each ice sheet individually to produce the normalized maps of sea level change in Fig. 4, the reader can consider scenarios that do not include NH ice sheets (NAIS and EIS) and only consider the traditional sources of Pliocene ice melt (EAIS, WAIS and GrIS; as we do, as case studies, in Fig. 6a and 7a).

Line 250. The age model for Enewetak Atoll has always been very uncertain. Table 3 of Wardlaw and Quinn paper is very hard to understand. The "mid-Pliocene" 3.6-3.5 Ma range of RSL change is -33m to +25m (extreme). This is not an equivalent interval to the M2-KM3.

Miller et al. (2012) cites the Enewetak Atoll backstripped record as having "peak sea level values among the three backstripped records are similar in the interval between 2.7 and 3.2 Ma (10-18 m in Virginia, 15-20 m in New Zealand, 20-25 m in Enewetak)." The age model is poorly constrained (3.0 + 0.5 Ma) and is averaged with 2.99 Ma peak observed in astronomically dated proxies.

But we emphasize that our study does not focus on peak sea level (and makes no assertion regarding the accuracy of estimates of this peak) but rather on the sea-level change across the MIS M2-KM3 stages of the mid-Pliocene and we merely use Enewetak Atoll as an illustrative site since there has been no published estimate of this sea-level change. Our maps indicate that if this sea-level change is constrained in the future it would overestimate GMSL$_P$ by ~10%. This is one illustrative example – the reader can make the same assessment of the difference between the local sea-level change and GMSL$_P$ for any site of interest given the results in the manuscript.

If I understand correctly this paper is modelling for the interval M2 to KM3 which is 3.3-3.15 Ma. In Grant et al P*lioSeaNZ* record this represents a change from +3 to +23m ESL (an overall increase of 20m over 8 eccentricity-modulated precision cycles). However, it should be noted that the PlioSeaNZ record floats, and while it constrains the amplitude and timing of ESL during G-I variability, it is only registered to present day sea-level through an assumption (see final paragraph). This paper would be greatly improved by the inclusion of a table showing the data used for amplitude and age of sea-level changes in the proxy records (e.g. Whanganui Basin, Enewetak Atoll).

We agree, but only to a limited extent. Our paper provides global maps of normalized sea-level changes across the MIS M2 to KM3 interval associated with various ice melt scenarios. Our aim is to provide any reader with the ability to assess the possible bias introduced by assuming that a local sea change across this interval faithfully records the global mean sea-level change, and we present some illustrative examples to emphasize the power of using these normalized maps for this purpose. We do not make assertions regarding the validity of any previous estimates of local sea-level changes across the interval and we will revise the manuscript to ensure that this point is emphasized. In this context, including information regarding data collected at Enewetak Atoll is not relevant to our study since there has been no published estimate of the sea-level change at

this site across the MIS M2 to KM3 interval. In contrast, we do use the one site where such an estimate has been made – Whanganui Basin (by the reviewer and colleagues!) – and we will include more information regarding the observations that underpin this estimate.

However, we emphasize again, that any mapping between the local sea-level change at this site and GMSL$_P$ that one can assess using our normalized maps – i.e., the 20% difference noted in our abstract – will *not depend on the observed sea-level change*. Placing too much emphasis on the observations may undermine this important point. Our revised text will make this issue far clearer.

Line 290, The authors claim a 20% underestimation in GMSL  (should be eustatic sea-level as stated above) from the Whanganui Basin, PlioSeaNZ record, which is unsurprisingly consistent with their ice sheet histories. However they should note,  that Grant et al. 2019 used a series of quite different hypothetical  ice sheet histories in their GIA modelling, which were based on best estimates from geological reconstructions, whereby +20 m of ESL was released under 4 scenarios **1**), 20 m ESL released from AIS only. **2**, AIS and GIS synchronously release 15 m and 5 m ESL, respectively **3**, AIS releases 25 m ESL while GIS accumulates 5 m ESL (that is, in antiphase). **4**, AIS and NHIS synchronously release 10 m ESL. In all cases the Whanganui Basin record lay geographically on the eustatic.

Any GIA model with an ice sheet history releasing 70% of the melt water from the northern hemisphere (such as in this paper) will overestimate the Whanganui record. Grant et al 2019 would have done the same had they used the ice sheet history used in Table 1 of this paper, which requires a total of 65m SLE ice is being melted between M2 and KM3. This ice sheet history is not supported by the published geological constraints. For example the total from Antarctic sector loss is greater than the total ice volume currently held in all the marine-based sectors and would require M2 glacial to be larger than present day ice volume, as well as having 45m SLE ice on the northern hemisphere continents.

We must emphasize that our manuscript does not argue for the accuracy of any specific melt scenario, and we will revise the text to ensure that this point is made clear. Table 1 provides the melt from each ice sheet in our various scenarios. These maximum melt scenarios were combined with the reference earth model and the calculate sea level was normalized using GMSL$_P$ to produce the maps in Fig. 4. As we demonstrated above, and will demonstrate in the revised text, these normalized maps can be used to determine what the local sea-level change will be for any net mass flux from each individual ice sheet. As example, per meter of GMSL$_P$ change, melt from North America across MIS stages M2 to KM3 will lead to 0.88 m of sea level change at Whanganui Basin (see Table 1 and Fig. 5). Providing the community with this information is important so that they can test *any scenario* for NAIS flux – or, indeed, flux from any other geographic region – against a future observation *at any site*. This utility is highlighted in our own discussion section where we considered five different melt scenarios that would produce 15 m of LSL change in the Whanganui Basin. These scenarios considered contributions to GMSL$_P$ from NAIS that ranged from 0 m to 9 m. We should also point out that melting from all 8 zones considered in Fig. 5 yielded a local sea level change across the MIS M2 to KM3 interval that was lower than the associated GMSL$_P$ of the melt – not just the NAIS melt – and this consistency is an important result. So, regardless of the mass flux scenario – including the 4

scenarios the reviewer mentions from the Grant et al. (2019) study – the local sea level change at Whanganui Basin will be lower than $GMSL_P$.

---

## Author Comment (AC2)

Our manuscript titled "The geometry of sea-level change across a mid-Pliocene glacial cycle" received two largely positive reviews that listed several suggestions for improvement. In the material below, we respond to the second comment. The reviewer comments appear in blue, and our responses are in black, with text quoted from the manuscript indented.

**Reviewer #2 (Anonymous):**

Thank you for the insight on our manuscript, our response to the following comments will be in bold.

In this manuscript, the Authors set up a GIA model for the deglaciation during the Mid-Pliocene Warm Period in order to explore the fingerprints of sea level change and, more specifically, their regional deviations from GMSL. In my opinion, this is an original and interesting contribution, the approach is technically sound and the manuscript is very well written, so I definitely support its publication. I have just a few comments for the Authors, which I hope will be useful to further improve the manuscript.

We thank the reviewer for these positive comments and their constructive review.

Lines 100-110: The definition of GMSL is a key point for this analysis, as the Authors remark in the paper. The definition that is adopted here (GMSLp) essentially takes into account only the ocean volume change in regions which were not covered by grounded ice at the beginning of the melt, in contrast with the more standard definition (GMSLs) which takes into account all the meltwater input. I think that it would be useful to better discuss the implications of the two definitions, and why GMSLp is the best choice for the present analysis.

We agree that our manuscript does not sufficiently motivate our adoption of the global mean sea level definition $GMSL_P$ – over other possible choices - in the normalization of the sea level calculations and we will revise the manuscript to address this issue. There are several important points to make in this regard:

- As long as a fully gravitationally self-consistent sea level theory is adopted (e.g., Kendall et al., 2005) the choice of GMSL definition will not impact the patterns evident in Fig. 4, only the numbers associated with the scale bar on that figure. Thus, the main aim of the paper – to explore "geometries of sea-level change" remains robust.
- However, in considering how geological observations at some sites relate to GMSL, the definition adopted for GMSL matters.
- Why do we believe that $GMSL_P$ is the most appropriate definition? In some sense, the strongest argument for this choice is evident in Fig. 4, where all the normalized sea level predictions lie very close to unity (1.0) on the equator. That is, in the far field of ice sheets one would expect that calculations over a long time window (in our case, from 3.295 Ma to 3.155 Ma) should be close to the global mean. This argument accords with the results in Pan et al. (2021) who showed that the outflux of water from exposed marine based sectors – which is included in the definition of $GMSL_P$ - tends to compensate for ice age (deformational and gravitational) dynamics. We can put this another way. If we had instead

adopted GMSL$_S$ as our definition then, for example, the normalized plot for the West
Antarctic scenario in Fig. 4 would have a value close to 0.6 (GMSL$_P$/GMSL$_S$ = 2.74/4.87).

- In the revised manuscript we will justify our choice of GMSL$_P$ more fully. In addition, we
  will add to the Discussion section a sub-section in which we more carefully compare this
  choice with other possible definitions for global mean sea level, including not only GMSL$_S$
  (which uniformly spreads total ice volume change over the global oceans) but also a
  definition based on uniformly spreading ice above floatation at 3.295 Ma over the ocean.

**Lines 169-171: Do the ice sheets have a constant thickness and a variable area?**

The ice sheets do not have a constant thickness. As per the model output from Berends et al.
(2019) the ice sheets are tied to modeled Pliocene topography, and therefore have varying
thicknesses and geographic extents through the MIS M2 to KM3 period. This can be seen in Fig.
3 where the ice geometries show both variable geographic extents (that contour to the continent
and change as ice melts) as well as variable thicknesses across the ice sheet extent (most notable
in the marine-based sectors).

**Lines 226-231: How the 24 models are generated? The lithospheric thickness and UM/LM
viscosities are randomly sampled in the given ranges or the ranges are scanned uniformly for
each of the three parameters?**

The 24 models were combinations of the following lithospheric thicknesses (72, 96 and 125 km),
and upper (0.2, 0.5 and 0.8 Pa s) lower mantle viscosities (5, 10, and 20 Pa s). Three models with
lithospheric thickness of 96 km and upper mantle viscosity of 0.8 Pa s were not included. All
earth models are within the range of models inferred from studies of GIA datasets (Mitrovica and
Forte, 2004; Lambeck et al., 2014). We will update the text to include this and add the three
model runs that were not included.

**Lines 291-297: it is not clear how the melt scenarios shown in Fig 7 are identified; a few more
details would be useful to fully understand this analysis.**

We agree with the reviewer that the rationale for these melt scenarios is vague. These five
scenarios were chosen to represent (a) commonly accepted sources of Pliocene ice sheet melt, as
well as (b) scenarios that encompass melt from only the Northern Hemisphere ice sheets, (c) ice
sheet contributions excluding East Antarctica (b and d), and (e) a scenario that includes melt
from all ice sheets. We will update the text to reflect this explanation, but we will also emphasize
that these choices were simply chosen to highlight possible departures of site-specific
observations from global mean sea level.

**Line 139: perhaps "3.89" should be "2.89"?**

This will be corrected.

**Lines 165-168 (and elsewhere): values of $\delta^{18}O$ are sometimes given without units and sometimes
with units of ‰; the notation should be made uniform.**

This will be corrected.

Figures 4 and 6: the three dots marking the positions of the sites discussed in the text are barely visible; I suggest using a larger and/or different symbol. Also, it could be useful to add in one of panels of Fig 4 three labels to help the reader identify the three locations.

This will be corrected.

Figure 5: the black circle in the box-and-whisker plot is hardly visible, also in this case I suggest using a larger/different symbol. Also the black line corresponding to the median is hard to see in the case of Enewetak Atoll.

This will be corrected.

---

## Author Response (AR1)

Our manuscript titled "The geometry of sea-level change across a mid-Pliocene glacial cycle" received two largely positive reviews that listed several suggestions for improvement. Below we respond to every comment, The reviewer comments appear in blue and our responses are in black. Text quoted from the manuscript is indented.

A few general comments regarding the revision are in order. First, in the original manuscript our calculations of GMSL$_S$ used the original "raw" ice volume changes in the eight ice sheet scenarios we considered. However, our sea level calculations make a check to ensure that all marine based ice is grounded, and they remove any ice that is determined to be floating. We should have computed GMSL$_S$ using the ice volumes after all floating ice was removed. This change led to minor corrections to the calculated values of GMSL$_S$ in Table 1, though we note that the values for the East and West Antarctic simulations were reduced by ~1 m of GMSL equivalent. Second, it is important in GIA sea level calculations that an iteration is performed to ensure that at the end of the calculation the present-day topography generated by the simulation matches the observed present-day topography. Our calculations of GMSL$_P$, which require knowledge of bedrock topography since they involve meltwater flux outside marine based areas, have been computed in the revised text using a stricter procedure that now ensures that the iterative process has fully converged. This change revised our calculated GMSL$_P$ values in Table, and thus our normalized maps (Fig. 4), by up to ~7%.

**Reviewer #1 (Dr. Naish):**

This paper is a nice model approach using state of the art GIA model to reconstruct GRD effects on global sea-level change during the time interval spanning the MIS M2 glacial to MIS KM3 interglacial as defined in the L&R05 d180 stack. Even though I have outlined below, why I believe the methodology is flawed, I would like to encourage the authors to consider trying a different range of ice sheet histories that might better reconcile with the far-field geological record of sea-level chnage. Its always difficult using a GIA model to evaluate a sea-level record when the ice sheet history is ambiguous.

I will declare up front that I am Tim Naish, and have been closely associated with the development of Whanganui Basin, NZ sea-level records. I also saw Meghan King present this paper at Fall AGU on 2022, where I discussed it briefly with her afterwards. I remian supportive of her work. I dont feel conflicted, but will leave that up to teh eds to decide.

It might help also if I mention the motivation behind the 2019 Grant et al study published in Nature. We were well-aware that the L&R05 d18O stack was of lower qulaity between 3.3-3 Ma due to low number of records and poor resolution of some of the records. The shallow marine glacial-interglacial sedimentary cycles in Whanganui are well dated in this interval as both Kaena and Mammoth paleomagnetic subchrons could be indentified as well as radiometrically-dated tephra. We argued for 2kyr sample resolution with an accuracy of +/- 4-5kyr at pmag transitions. On this basis we built an independently dated sea-level record and showed that it was largely in phase with Antarctic summer insolation. Given geological evidence precluding a large NH ice sheet until 2.7Ma (Haug et al., 2005; Jansen et al., 2000; Brigham-Grette et al., 2013; Berends et al., 2019, Eldrett et al., 2007; Thiede et al., 2011; Tripati & Darby, 2018). we also

argued most of the meltwater was of Antarctic origin, and this informed the GIA modeling to we used to test how close Whanganui RSL was to ESL.

My comments follow.

We thank Dr. Naish for his review and constructive comments.

Line 100. Definition of global mean sea-level needs to be registered to the centre of the Earth, otherwise it is eustatic sea-level. Im OK with Pan et al 2022 method for estimating change in eustatic sea-level ESL (but shouldnt use GMSL unless you can register it to present day sea-level).

In studies of paleo sea level, GMSL changes are not referenced to the center of the Earth. Any change in the elevation of a geological marker reflects *relative* sea level change – that is, the change in the sea surface height relative to any crustal elevation change. In modern sea level studies, tide gauge records also reflect the change in the distance between the sea surface and crust, while satellite records measure "*absolute* sea level" changes, i.e., changes in the sea surface height relative to some reference frame (e.g., the center of the Earth).

In the geological literature of long term sea-level change there has been an argument that only the change in the sea surface height, or absolute sea level, is important (e.g., http://stratigrafia.org/sequence/accommodation.html) but this doesn't hold if there is any departure at all from hydrostatic (not just isostatic) equilibrium. In an ice age calculation, there is always such a departure because of the adoption of an elastic lithosphere and unrelaxed viscous stresses.

Perhaps the issue arises from the fact that our text does not always make clear that we are usually speaking of GMSL *changes*. To address this, we have revised the text to ensure that all mentions of GMSL are connected to the word "change" where appropriate. For example, on line 489:

> The two maps identify geographic regions in which the LSL variation might provide the closest measure of GMSL change from MIS M2 to KM3.

Furthermore, we are explicit in defining $GMSL_P$ and $GMSL_S$ as the GMSL *change* computed using the two definitions discussed in the text.

Finally, we note that the terms "eustatic" and "eustasy" are being avoided because of the ambiguity associated with their definition (Gregory, et al. "Concepts and terminology for sea level: Mean, variability and change, both local and global." *Surveys in Geophysics, 40,* 1251-1289, 2019).

Line 145. The Brerends et al. ice sheet histories puts more ice on northern hemisphere continents during M2 than geological evidence implies (see above refs).

Yes, the Berends et al. (2019) ice sheet histories certainly suggest significantly more Northern Hemisphere (i.e., NAIS and EIS) ice than other studies. The reviewer's comment does, however,

point to another way in which our manuscript can be improved. The normalization procedure we apply to the raw sea level calculation to yield Fig. 4 allows us in principle to scale these figures by any global mean sea level scenario since the sea level calculations are (quasi) linearly related to the total mass flux. Therefore, the normalized sea level map for the NAIS scenario is not very sensitive to a plausible range in the GMSL change assumed for the scenario.

As an illustration of this, we have also added Figure S1 to the Supplementary Material (lines 971-974) that compares the normalized sea level map of the NAIS scenario from Figure 4 of the manuscript (GMSL$_P$ = 32.95 m) with the equivalent map for an alternate NAIS scenario characterized by GMSL$_P$ = 7.71 m. The two maps are nearly identical in areas away from the ancient ice cover. So, in this sense, even if the 33 m scenario is considered unrealistic, the normalized sea level change holds for any other scenario, at least for sites away from the ice cover.

[Figure]

**Figure S1.** Comparison of normalized sea level maps for MIS M2 to KM3 NAIS collapse where GMSL$_P$ is (a) 32.95 m and (b) 7.71 m. Predictions are based on the reference viscoelastic Earth model described in the text, and the three black dots on the figure show the location of continental shelf/upper slope sites discussed in the text.

The revised text includes two new passages which read (lines 232-240):

> Finally, the global maps of sea-level change calculated for each ice melt scenario are normalized by the GMSL$_P$ value associated with that scenario (Table 1). Since the sea level predictions are quasi-linearly related to the net ice mass flux, this normalization procedure yields maps that are - outside the immediate vicinity of the melt zone - relatively insensitive to the GMSL change, or equivalently the total ice mass flux, of the scenario. We demonstrate this insensitivity in the results below. The linearity also allows one to combine, with suitable weighting, the maps for individual melt zones, to assess the connection between LSL change at any site and total GMSL$_P$ for any scenario of interest. This generality is an important point to emphasize because we make no assertion regarding the validity of the total melt volumes in each of the eight scenarios listed in Table 1, and our main conclusions regarding biases in the mapping between local and global sea level are insensitive to these melt volumes.

And (line 310-315):

> As discussed in the Introduction, the normalization procedure applied in each scenario within Figure 4 should yield maps that are relatively insensitive to changes in the net volume of melt if the geometry of the ice melt is not fundamentally altered. To highlight this issue,

Supplementary Figure 1 (Fig. S1) shows a map analogous to the NAIS scenario in Figure 4 with the exception that we adopted a melt model with a $GMSL_P$ value of 7.71 m. Outside of the region in the near vicinity of the mass flux, the two normalized maps show nearly identical structure. Of course, the sensitivity is larger at sites close to the mass flux, as we discuss below.

Line 160. Uses  the assumption that L&R05 is valid time variation in global ice volume for this time interval. Other studies have raised concerns about the frequency and amplitude of the stack between 3.3-3 Ma, where d18O records are low in number and resolution, leading to potential artefacts through the stacking process (Patterson et al., 2014, Nature Geo; Grant & Naish, 2021, Pages). Note that the highest resolution d18O record during this time interval is dominated be precession (ODP 846-849), as it should be due to a node in obliquity in the orbital solution. Note also the Grant et al 2019 sea-level record which is independent of the L&R05 stack correlates strongly with high southern latitude insolation dominated by precession.

Whereas the last comment dealt with the net magnitude of the GMSL change between MIS M2 to KM3, this comment is focused on potential inaccuracies in the details of the sea-level oscillations between these two times as mapped from the d18O stack of Lisiecki and Raymo.  To consider this issue, we performed a series of calculations in which we varied the magnitude of the multiple smaller sea level oscillations between MIS M2 and KM3 - and the total time difference between these stages. These additional simulations produced negligible impacts on the normalized sea level maps. Accordingly, we added the following sentence to the manuscript (lines 315-317):

> Additionally, the sensitivity analyses with a 120 kyr time duration and smaller sea-level oscillations between MIS M2 and KM3 (Fig. 2) revealed that the normalized sea level maps were negligibly impacted.

Furthermore, the text has been revised to discuss the uncertainties in oxygen isotope records and to introduce the additional simulations (lines 242-260).

> With respect to the normalized $\delta^{18}O$ time series utilized in this study, there are uncertainties in the LR04 stack derived frequency and amplitude of 3.3-3 Ma glacial-interglacial cycles. The stack was put together from 57 different benthic $\delta^{18}O_{carb}$ and Mg/Ca ratios (Lisiecki and Raymo, 2005), and is complicated by uncertainties in fossil species and proxy specific calibrations, alteration due to diagenesis, and changes in seawater chemistry (Raymo et al., 2017). Additionally, studies of ice-berg rafted debris from areas proximal to the EAIS suggest that, unlike $\delta^{18}O$ records over the 3.3-3 Ma time period, glacial-interglacial cycles were not paced by obliquity (40 kyr) but instead (23 kyr) precession (Patterson et al., 2014). Therefore, to accommodate these uncertainties we performed sensitivity analyses in which we shortened the time duration between MIS M2 and KM3 from 140 kyr to 120 kyr or randomly perturbed the magnitude of the smaller sea-level oscillations between the two marine isotope stages (Fig. 2).

Note also that the proxy ice sheet histories for this time interval are not well constrained by d18O or the Berends modelling, or the ice berg rafted debris records from Antarctica and the Arctic (this should be brought into the discussion). Certainly high latitude northern hemisphere IBRD records (N Pacific, Norwegian Sea, E Greenland) show no continental scale NH ice sheet margins (as implied by Berends et al), with the exception of Greenland. This is why there should be caution taken rather just than accepting Berends et al., as a "series of Pliocene-realistic ice geometries" (line 144).

The new material (and Fig. S1) discussed above now emphasizes that uncertainties associated with the size of Pliocene NH ice sheets will not map into large uncertainties in the the normalized sea level-change maps. Nevertheless, we emphasize that since we model each ice sheet individually to produce the normalized maps of sea level change in Figure 4, the reader can consider scenarios that do not include NH ice sheets (NAIS and EIS) and only consider the traditional sources of Pliocene ice melt (EAIS, WAIS and GrIS; as we do, as case studies, in Fig. 6a and 7a). We have updated the manuscript text to make this point clearer (line 485):

(One can repeat the same exercise with any weighted combination of the maps in Fig. 4.)

Line 250. The age model for Enewetak Atoll has always been very uncertain. Table 3 of Wardlaw and Quinn paper is very hard to understand. The "mid-Pliocene" 3.6-3.5 Ma range of RSL change is -33m to +25m (extreme). This is not an equivalent interval to the M2-KM3.

Miller et al. (2012) describes the Enewetak Atoll backstripped record as having "peak sea level values among the three backstripped records are similar in the interval between 2.7 and 3.2 Ma (10-18 m in Virginia, 15-20 m in New Zealand, 20-25 m in Enewetak)." The age model is poorly constrained (3.0 + 0.5 Ma) and is averaged with 2.99 Ma peak observed in astronomically dated proxies.

Yet, we emphasize that our study does not focus on constraining peak sea level (and makes no assertion regarding the accuracy of estimates of this peak), but rather focuses on the sea-level change across the MIS M2-KM3 stages. We merely use Enewetak Atoll as an illustrative site since there is a Pliocene age record here—albeit imperfectly dated—yet there has been no published estimate of the amplitude of sea-level change across the MIS M2-KM3 stages. Our maps indicate that if this sea-level change is constrained in the future it would overestimate $GMSL_P$ by ~10%. To clarify that the reader can make the same assessment of the difference between the local sea-level change and $GMSL_P$ for any site of interest given the results in the manuscript we have added this as a discussion point. For example, the revised text on lines 478-480 reads:

We emphasize that these three sites are chosen as illustrative case studies, and that the maps in Figure 4 can be used to assess the relationship between LSL and $GMSL_P$ for any site and for any of the eight melt scenarios.

If I understand correctly this paper is modelling for the interval M2 to KM3 which is 3.3-3.15 Ma. In Grant et al P*lioSeaNZ* record this represents a change from +3 to +23m ESL (an overall increase of 20m over 8 eccentricity-modulated precision cycles). However, it should be noted

that the PlioSeaNZ record floats, and while it constrains the amplitude and timing of ESL during G-I variability, it is only registered to present day sea-level through an assumption (see final paragraph). This paper would be greatly improved by the inclusion of a table showing the data used for amplitude and age of sea-level changes in the proxy records (e.g. Whanganui Basin, Enewetak Atoll).

Our paper provides global maps of normalized sea-level changes across the MIS M2 to KM3 interval associated with various ice melt scenarios. Our aim is to provide any reader with the ability to assess the possible bias introduced by assuming that a local sea change across this interval faithfully records the global mean sea-level change, and we present some illustrative examples to emphasize the power of using these normalized maps for this purpose. We do not make assertions regarding the validity of any previous estimates of local sea-level changes across the interval and have revised the manuscript to ensure that this point is emphasized. For example (line 238-240):

> …we make no assertion regarding the validity of the total melt volumes in each of the eight scenarios listed in Table 1, and our main conclusions regarding biases in the mapping between local and global sea level are insensitive to these melt volumes.

In this context, including information regarding data collected at Enewetak Atoll is not relevant to our study since there has been no published estimate of the sea-level change at this site across the MIS M2 to KM3 interval. In contrast, we do use the one site where such an estimate has been made – Whanganui Basin (by the reviewer and colleagues) – and in the revised manuscript text we have included more information regarding the observations that underpin this estimate. In lines 595-614 we discuss alternative definitions of GMSL, of which $GMSL_{IAF}$ – or ice above flotation – is more in line with the assumptions of Grant et al. (2019).

> Other definitions of GMSL change are, of course, possible. Figure S2 extends Figure 1 to include two other possibilities. The first, $GMSL_{IAF}$, involves spreading the ice volume above floatation as defined at the start of MIS M2 over the global ocean. This ignores the flux of water from exposed marine sectors which will be a significant limitation considering the time duration of the MPWP interval we are considering (~140 kyr) in scenarios with considerable ice sheet retreat from such sectors. The second, $GMSL_S$, takes the full volume of meltwater between MIS M2 and KM3 and spreads it over the ocean. As in the case of $GMSL_P$, the area of ocean used in the calculation (i.e., whether or not the marine sector is included) will have ~1% or less effect on $GMSL_S$. One can interpret $GMSL_S$ as a special case of $GMSL_P$ in which any exposed marine based sectors rebound sufficiently in the calculation of $GMSL_P$ that they become subaerial. This will, of course, depend on the volume of the marine accommodation space relative to the total post-glacial uplift of the crust from MIS M2 to KM3. Table 1 also shows the $GMSL_S$ value computed for each ice history described above. The limitation of adopting this definition is most pronounced in the results for West Antarctica, where substantial marine-based regions are exposed across the ice history. The difference in the GMSL calculations (4.07 - 2.92 ~ 1.15 m) largely reflects, in the calculation of $GMSL_P$, the volume of meltwater that remains in these marine-based sectors at MIS KM3 that were exposed by grounded ice retreat from MIS M2 to KM3. If one were to use $GMSL_S$ instead of $GMSL_P$, then the normalized map of the WAIS scenario in Figure 4 would show

values of ~0.7 (2.92/4.07) rather than 1.0 near the equator, i.e., the "far field", which suggests that $GMSL_S$ is not an appropriate metric for GMSL change in this case. The metric $GMSL_P$ yields values intermediate to $GMSL_{IAF}$ and $GMSL_S$ and all three definitions of GMSL change will be identical in the case where no grounded, marine-based ice is involved in an ice melt scenario. The latter is close to being the case in the GrIS scenario we have adopted.

However, we emphasize again, that any mapping between the local sea-level change at this site and $GMSL_P$ that one can assess using our normalized maps – i.e., the 15% difference noted in our abstract – will *not depend on the observed sea-level change*. Placing too much emphasis on the observations may undermine this important point.

Line 290, The authors claim a 20% underestimation in GMSL (should be eustatic sea-level as stated above) from the Whanganui Basin, PlioSeaNZ record, which is unsurprisingly consistent with their ice sheet histories. However they should note, that Grant et al. 2019 used a series of quite different hypothetical ice sheet histories in their GIA modelling, which were based on best estimates from geological reconstructions, whereby +20 m of ESL was released under 4 scenarios **1**), 20 m ESL released from AIS only. **2**, AIS and GIS synchronously release 15 m and 5 m ESL, respectively **3**, AIS releases 25 m ESL while GIS accumulates 5 m ESL (that is, in antiphase). **4**, AIS and NHIS synchronously release 10 m ESL. In all cases the Whanganui Basin record lay geographically on the eustatic.

Any GIA model with an ice sheet history releasing 70% of the melt water from the northern hemisphere (such as in this paper) will overestimate the Whanganui record. Grant et al 2019 would have done the same had they used the ice sheet history used in Table 1 of this paper, which requires a total of 65m SLE ice is being melted between M2 and KM3. This ice sheet history is not supported by the published geological constraints. For example the total from Antarctic sector loss is greater than the total ice volume currently held in all the marine-based sectors and would require M2 glacial to be larger than present day ice volume, as well as having 45m SLE ice on the northern hemisphere continents.

Table 1 provides the melt from each ice sheet in our various scenarios. These maximum melt scenarios were combined with the reference earth model and the calculate sea level was normalized using $GMSL_P$ to produce the maps in Figure 4. As we demonstrated above, these normalized maps can be used to determine what the local sea-level change will be for any net mass flux from each individual ice sheet. As we have noted above, the manuscript has been revised at several points to emphasize the generality of the viscoelastic fingerprints, including with the addition of the new Fig. S1.

As example, per meter of $GMSL_P$ change, melt from North America across MIS stages M2 to KM3 will lead to 0.89 m of sea level change at Whanganui Basin (see Table 1 and Fig. 5). Providing the community with this information is important so that they can test *any scenario* for NAIS flux – or, indeed, flux from any other geographic region – against a future observation *at any site*. This utility is highlighted in our own discussion section where we considered five different melt scenarios that would produce 15 m of LSL change in the Whanganui Basin. These scenarios considered contributions to $GMSL_P$ from NAIS that ranged from 0 m to 9.5 m. We

should also point out that melting from all 8 zones considered in Figure 5 yielded a LSL change across the MIS M2 to KM3 interval that was lower than the associated $GMSL_P$ of the melt – not just the NAIS melt – and this consistency is an important result. *So, regardless of the mass flux scenario – including the 4 scenarios the reviewer mentions from the Grant et al. (2019) study – the LSL change at Whanganui Basin will be lower than $GMSL_P$.*

**Reviewer #2 (Anonymous):**

In this manuscript, the Authors set up a GIA model for the deglaciation during the Mid-Pliocene Warm Period in order to explore the fingerprints of sea level change and, more specifically, their regional deviations from GMSL. In my opinion, this is an original and interesting contribution, the approach is technically sound and the manuscript is very well written, so I definitely support its publication. I have just a few comments for the Authors, which I hope will be useful to further improve the manuscript.

We thank the reviewer for these positive comments and their constructive review.

Lines 100-110: The definition of GMSL is a key point for this analysis, as the Authors remark in the paper. The definition that is adopted here (GMSLp) essentially takes into account only the ocean volume change in regions which were not covered by grounded ice at the beginning of the melt, in contrast with the more standard definition (GMSLs) which takes into account all the meltwater input. I think that it would be useful to better discuss the implications of the two definitions, and why GMSLp is the best choice for the present analysis.

We agree that our manuscript does not sufficiently motivate our adoption of the global mean sea level definition $GMSL_P$ – over other possible choices - in the normalization of the sea level calculations and we have revised the manuscript to address this issue. There are several important points to make in this regard:

- As long as a fully gravitationally self-consistent sea level theory is adopted (e.g., Kendall et al., 2005) the choice of GMSL definition will not impact the patterns evident in Figure 4, only the numbers associated with the scale bar on that figure. Thus, the main aim of the paper – to explore "geometries of sea-level change" remains robust.
- However, in considering how geological observations at some sites relate to GMSL, the definition adopted for GMSL matters.
- Why do we believe that $GMSL_P$ is the most appropriate definition? In some sense, the strongest argument for this choice is evident in Figure 4, where the normalized sea level predictions lie close to unity (1.0) on the equator. That is, in the far field of ice sheets one would expect that calculations over a long time window (in our case, from 3.295 Ma to 3.155 Ma) should be close to the global mean. This argument accords with the results in Pan et al. (2021) who showed that the outflux of water from exposed marine based sectors –

which is included in the definition of $GMSL_P$ - tends to compensate for ice age (deformational and gravitational) dynamics. We can put this another way. If we had instead adopted $GMSL_S$ as our definition then, for example, the normalized plot for the West Antarctic scenario in Figure 4 would have a value close to 0.7 ($GMSL_P/GMSL_S$ = 2.92/4.07).

We have revised the manuscript text to justify our choice of $GMSL_P$ more fully (lines 586-593). We have pushed the discussion of alternative GMSL definitions to the Discussion section, and to avoid confusion only refer to $GMSL_P$ in the Methods and Results.

> The definition we have adopted, $GMSL_P$, involves dividing the total meltwater volume that enters the open ocean outside any exposed marine based sectors from MIS M2 to KM3 by the area of the ocean (Fig. 1). The appropriateness of this choice is suggested by the normalized sea-level change maps of Figure 4, which are all characterized by values within a few percent of 1.0 along the equator. (Mean equatorial ocean values are: Eurasia: 0.9870, Greenland: 0.9695, North America: 0.9703, West Antarctica: 1.0150, East Antarctica: 1.0000, Aurora Basin: 0.9807, Prydz Bay: 0.9835, and Wilkes Basin: 0.9717 for the normalized maps in Fig. 4.) That is, at sites furthest afield from the deformational, gravitational and rotational effects of the GIA, the calculated sea-level change reflects the GMSL change.

In the Discussion section we more carefully compare this choice with other possible definitions for GMSL, including not only $GMSL_S$ (which uniformly spreads total ice volume change over the global oceans) but also $GMSL_{IAF}$, a definition based on uniformly spreading ice above floatation at 3.295 Ma over the ocean. These issues are discussed in the following text (line 595-614):

> Other definitions of GMSL change are, of course, possible. Figure S2 extends Figure 1 to include two other possibilities. The first, $GMSL_{IAF}$, involves spreading the ice volume above floatation as defined at the start of MIS M2 over the global ocean. This ignores the flux of water from exposed marine sectors which will be a significant limitation considering the time duration of the MPWP interval we are considering (~140 kyr) in scenarios with considerable ice sheet retreat from such sectors. The second, $GMSL_S$, takes the full volume of meltwater between MIS M2 and KM3 and spreads it over the ocean. As in the case of $GMSL_P$, the area of ocean used in the calculation (i.e., whether or not the marine sector is included) will have ~1% or less effect on $GMSL_S$. One can interpret $GMSL_S$ as a special case of $GMSL_P$ in which any exposed marine based sectors rebound sufficiently in the calculation of $GMSL_P$ that they become subaerial. This will, of course, depend on the volume of the marine accommodation space relative to the total post-glacial uplift of the crust from MIS M2 to KM3. Table 1 also shows the $GMSL_S$ value computed for each ice history described above. The limitation of adopting this definition is most pronounced in the results for West Antarctica, where substantial marine-based regions are exposed across the ice history. The difference in the GMSL calculations (4.07 - 2.92 ~ 1.15 m) largely reflects, in the calculation of $GMSL_P$, the volume of meltwater that remains in these marine-based sectors at MIS KM3 that were exposed by grounded ice retreat from MIS M2 to KM3. If one were to use $GMSL_S$ instead of $GMSL_P$, then the normalized map of the WAIS scenario in Figure 4 would show

values of ~0.7 (2.92/4.07) rather than 1.0 near the equator, i.e., the "far field", which suggests that GMSL$_S$ is not an appropriate metric for GMSL change in this case. The metric GMSL$_P$ yields values intermediate to GMSL$_{IAF}$ and GMSL$_S$ and all three definitions of GMSL change will be identical in the case where no grounded, marine-based ice is involved in an ice melt scenario. The latter is close to being the case in the GrIS scenario we have adopted.

Lines 169-171: Do the ice sheets have a constant thickness and a variable area?

The ice sheets do not have a constant thickness. As per the model output from Berends et al. (2019) the ice sheets are tied to modeled Pliocene topography, and therefore have varying thicknesses and geographic extents through the MIS M2 to KM3 period. This can be seen in Figure 3 where the ice geometries show both variable geographic extents (that contour to the continent and change as ice melts) as well as variable thicknesses across the ice sheet extent (most notable in the marine-based sectors).

Lines 226-231: How the 24 models are generated? The lithospheric thickness and UM/LM viscosities are randomly sampled in the given ranges or the ranges are scanned uniformly for each of the three parameters?

The 27 models were combinations of the following lithospheric thicknesses (72, 96 and 125 km), and upper (0.3, 0.5 and 0.8 Pa s) lower mantle viscosities (5, 10, and 20 Pa s). All earth models are within the range of models inferred from studies of GIA datasets (Mitrovica and Forte, 2004; Lambeck et al., 2014). The text has been updated to reflect this explanation (lines 111-113):

> However, we also perform an analysis that explores the sensitivity of the normalized sea level predictions to plausible variations in the viscosity model. These additional 27 models are combinations of the following lithospheric thicknesses (72, 96, and 125 km) and upper (0.3, 0.5, and 0.8 Pa s) and lower mantle viscosities (5, 10, and 20 Pa s).

Lines 291-297: it is not clear how the melt scenarios shown in Fig 7 are identified; a few more details would be useful to fully understand this analysis.

We agree with the reviewer that the rationale for these melt scenarios is vague. These five scenarios were chosen to represent one set of commonly accepted sources of Pliocene ice sheet melt (a), a scenario that excludes ice sheet contributions from North America (b) and East Antarctica (c), and two scenarios that includes melt from all ice sheets (d and e). These choices were simply chosen to highlight possible departures of site-specific observations from global mean sea level. The text has been updated to reflect this explanation (line: 544-546):

> The five scenarios presented in Figure 7 were chosen to represent one set of commonly accepted sources of Pliocene ice sheet melt (a), a scenario that excludes ice sheet contributions from North America (b) and East Antarctica (c), and two scenarios that includes melt from all ice sheets (d and e).

Line 139: perhaps "3.89" should be "2.89"?

We have corrected this typo.

Lines 165-168 (and elsewhere): values of $\delta^{18}O$ are sometimes given without units and sometimes with units of ‰; the notation should be made uniform.

We have corrected the inconsistent use of units.

Figures 4 and 6: the three dots marking the positions of the sites discussed in the text are barely visible; I suggest using a larger and/or different symbol. Also, it could be useful to add in one of panels of Fig 4 three labels to help the reader identify the three locations.

We have updated Figures 4-6 to increase the size of the black circle, and the first panel in each figure is labeled with the name of the location.

Figure 5: the black circle in the box-and-whisker plot is hardly visible, also in this case I suggest using a larger/different symbol. Also the black line corresponding to the median is hard to see in the case of Enewetak Atoll.

We have updated Figure 5 to increase the size of the black circle and adjust the color of the Enewetak Atoll box and whisker plots.